# LARP1 functions as a molecular switch for mTORC1-mediated translation of an essential class of mRNAs

Sungki Hong[1], Mallory A Freeberg[1,2†], Ting Han[1†‡], Avani Kamath[1†], Yao Yao[1], Tomoko Fukuda[1], Tsukasa Suzuki[1§], John K Kim[1,3¶], Ken Inoki[1,4,5*]

[1]Life Sciences Institute, University of Michigan, Ann Arbor, United States; [2]Department of Computational Medicine and Bioinformatics, University of Michigan Medical School, Ann Arbor, United States; [3]Department of Human Genetics, University of Michigan Medical School, Ann Arbor, United States; [4]Department of Molecular and Integrative Physiology, University of Michigan Medical School, Ann Arbor, United States; [5]Department of Internal Medicine, University of Michigan Medical School, Ann Arbor, United States

*For correspondence: inokik@ umich.edu

[†]These authors contributed equally to this work

Present address: [‡]Department of Biochemistry, UT Southwestern Medical Center, Dallas, United States; [§]Faculty of Applied Bioscience, Tokyo University of Agriculture, Tokyo, Japan; [¶]Department of Biology, Johns Hopkins University, Baltimore, United States

Competing interests: The authors declare that no competing interests exist.

**Abstract** The RNA binding protein, LARP1, has been proposed to function downstream of mTORC1 to regulate the translation of 5'TOP mRNAs such as those encoding ribosome proteins (RP). However, the roles of LARP1 in the translation of 5'TOP mRNAs are controversial and its regulatory roles in mTORC1-mediated translation remain unclear. Here we show that LARP1 is a direct substrate of mTORC1 and Akt/S6K1. Deep sequencing of LARP1-bound mRNAs reveal that non-phosphorylated LARP1 interacts with both 5' and 3'UTRs of RP mRNAs and inhibits their translation. Importantly, phosphorylation of LARP1 by mTORC1 and Akt/S6K1 dissociates it from 5'UTRs and relieves its inhibitory activity on RP mRNA translation. Concomitantly, phosphorylated LARP1 scaffolds mTORC1 on the 3'UTRs of translationally-competent RP mRNAs to facilitate mTORC1-dependent induction of translation initiation. Thus, in response to cellular mTOR activity, LARP1 serves as a phosphorylation-sensitive molecular switch for turning off or on RP mRNA translation and subsequent ribosome biogenesis.

## Introduction

Mechanistic target of rapamycin complex 1 (mTORC1) functions as a positive regulator of translation initiation and protein synthesis to promote cell growth and proliferation (*Bhat et al., 2015*; *Dibble and Manning, 2013*). Short-term treatment with rapamycin, an allosteric mTORC1 inhibitor, only partially inhibits global protein synthesis but effectively blocks the translation of certain 5' terminal oligopyrimidine tract (5'TOP) mRNAs (*Hinnebusch et al., 2016*; *Jefferies et al., 1997*; *Meyuhas and Kahan, 2015*). In contrast, recent studies using newly developed specific mTOR kinase inhibitors such as Torin1 demonstrate that complete inhibition of cellular mTOR kinase activity results in strong suppression of nearly all mRNA translation (*Hsieh et al., 2012*; *Thoreen et al., 2012*). However, the sensitivity of translation inhibition by mTOR kinase inhibitors still varies significantly among different mRNAs, and the translation of mRNAs containing pyrimidine-enriched sequence (PES) in their 5'UTRs (i.e., 5'TOP, TOP-like, and pyrimidine rich translation element (PRTE) sequences) is much more effectively inhibited. Moreover, the sensitivity of translation inhibition by mTOR inhibitors also varies within PES-containing mRNAs.

The 4EBP family of proteins have been proposed to play a key role in suppressing the translation of PES-containing mRNAs (*Thoreen et al., 2012*). However, the molecular mechanisms by which

inhibition of active eIF4F complex formation by 4EBPs further potentiates translation inhibition of PES-containing mRNAs remain elusive (*Miloslavski et al., 2014*). Recent studies demonstrate that La-related proteins 1 (LARP1), an evolutionarily conserved RNA binding protein, interacts with components of the active eIF4F complex and mTORC1 and regulates the translation of TOP mRNAs (*Tcherkezian et al., 2014*). LARP1 directly interacts with the TOP sequences of 5'TOP mRNAs such as those that encode ribosome proteins (RP) in vitro and stabilizes RP mRNAs in vivo (*Aoki et al., 2013*; *Fonseca et al., 2015*; *Lahr et al., 2015*). However, the roles of LARP1 in mTORC1-mediated RP mRNA translation remain controversial because previous studies propose conflicting models wherein LARP1 functions as either a positive or negative regulator of RP mRNA translation (*Fonseca et al., 2015*; *Tcherkezian et al., 2014*). Furthermore, how LARP1 involves in mTORC1-mediated RP mRNA translation also remains unclear.

In this report, we investigated the molecular mechanisms of LARP1 function in the mTORC1-mediated translation of RP mRNAs. We first identified mRNAs and sequences directly bound by endogenous LARP1 in vivo under normal growing and mTORC1-inhibited conditions using photoactivatable ribonucleoside–enhanced crosslinking and immunoprecipitation (PAR-CLIP) (*Hafner et al., 2010*). As predicted, LARP1 directly interacts with pyrimidine-enriched sequences (PES) of mRNAs such as RP mRNAs that significantly overlap with those regulated by mTOR activity. However, LARP1 interacts with the 3'UTR of RP mRNAs under growth conditions while it also binds to specific PES at the 3'end of their 5'UTRs when mTOR activity is inhibited. Thus, LARP1 may not be a bona fide 5'TOP binding protein in vivo. We identified that these dynamic LARP1 interactions with RP mRNAs are regulated through direct phosphorylations of LARP1 by mTORC1 and Akt/S6K1. Phosphorylation of LARP1 induces its dissociation from the PES in 5'UTRs but enhances its binding to 3'UTRs of RP mRNAs. Importantly, phosphorylated LARP1 also functions as a scaffolding protein for mTORC1 on translationally-competent LARP1-interacting mRNAs to facilitate mTORC1-dependent phosphorylation of its substrate proteins, 4EBP1 and S6K1, processes that are essential for translation initiation and elongation. Thus, the spatial recruitment of mTORC1 by LARP1 to specific translational machinery may provide significant advantages for the translation of LARP1-associated RP mRNAs. As a unique substrate of mTORC1 and Akt/S6K1, we propose that LARP1 functions as a phosphorylation-sensitive molecular switch in the translation of an essential class of mRNAs as well as an important regulator of mTORC1 itself.

## Results

### Dynamic LARP1 interaction with RP mRNAs in an mTOR activity-dependent manner

While several recent studies have indicated that LARP1 associates with 5'TOP mRNAs through their TOP sequences or polyA tails (*Aoki et al., 2013*; *Lahr et al., 2015*), the comprehensive identity and sequence characteristics of mRNAs that preferentially interact with LARP1 have not been defined. To address this gap, we performed PAR-CLIP of endogenous LARP1 in HEK293T cells in the presence or absence of an mTOR inhibitor (PP242), followed by deep sequencing of the LARP1-bound RNA substrates (The data set was deposited: GEO: GSE59599). One advantage of PAR-CLIP over conventional UV crosslinking methodologies is the signature of specific T-to-C conversions in the resulting sequencing reads that mark where the incorporated 4-thiouracil of RNAs form covalent linkages with the interacting protein (*Hafner et al., 2010*). Sequenced reads were mapped using Bowtie (*Langmead et al., 2009*) to the human transcriptome, clustered to derive LARP1 binding sites, and filtered to retain clusters containing 0–2 T-to-C conversion events and passing an empirically-derived reads per million mapped reads (RPM) threshold (details in Materials and methods; *Figure 1—figure supplement 1A*; *Supplementary file 1*). In parallel, replicate mRNA-seq experiments were performed in the presence or absence of PP242 to quantify gene expression and normalize LARP1 binding sites to mRNA abundance levels. We identified 1200 and 1,900 LARP1 binding sites on 1000 and 1500 mRNAs in the presence or absence of PP242, respectively (*Figure 1—figure supplement 1A and B*).

Gene ontology (GO) term enrichment analysis of genes bound by LARP1 in growing conditions revealed enrichment for terms related to translation. LARP1 was bound to 137 translation-related genes, including 42 genes encoding ribosomal proteins (RP), as well as genes involved in cellular

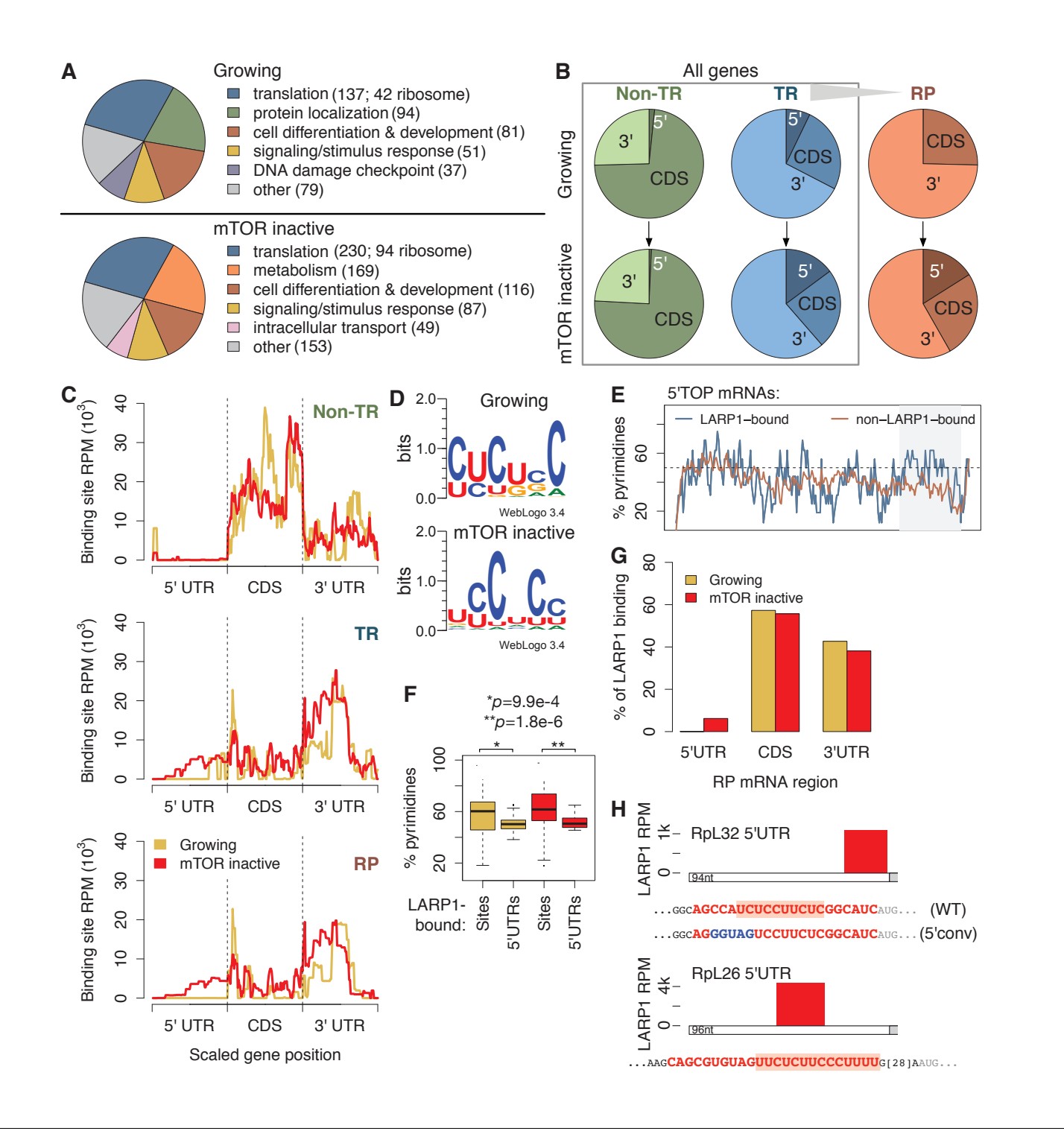

**Figure 1.** LARP1 binds pyrimidine-rich 5'UTR regions of translation-related transcripts. (**A**) LARP1-bound genes are most enriched for GO terms related to translation including RP genes. (**B**) Upon mTOR inactivation, LARP1 binding at 5'UTRs increases on TR and RP genes. (**C**) LARP1 binding at TR and RP 5'UTRs under mTOR-inactive conditions tends to occur at the 3' end. (**D**) LARP1 binds directly to pyrimidine-enriched sequences in 5'UTRs. (**E**) The LARP1 binding sites at the 3' end of 5'TOP-containing 5'UTRs are enriched for pyrimidines. (**F**) LARP1-bound sites on 5'UTRs are enriched for pyrimidines compared to the rest of the 5'UTR sequence. Welch's two-tailed *t*-test: *p=1.4e-15 and **p=1.2e-18. (**G**) LARP1 binding on RP-encoding mRNAs is gained at 5'UTRs upon mTOR inactivation and slightly decreased at CDSs and 3'UTRs. (**H**) The locations (red box), sequences (red color), and

*Figure 1 continued on next page*

*Figure 1 continued*

motifs (orange background) of the 5' UTR LARP1 binding site in RpL32 and RpL26 mRNAs under mTOR-inactive conditions. Substituted nucleotides are highlighted by blue color.

The following figure supplement is available for figure 1:

**Figure supplement 1.** LARP1 binds 5'UTR pyrimidine-rich regions of translation-related transcripts.

differentiation and development (*Figure 1A*; *Supplementary file 2*). Under mTOR-inactive conditions, more (230) translation-related genes were bound by LARP1, including 94 RP-encoding genes (*Figure 1A*; *Supplementary file 2*). These results indicate that LARP1 substrates are enriched for mRNAs encoding factors involved in translation, and that this interaction is enhanced under conditions of mTOR inactivation.

To identify where LARP1 binds across a transcript, we summed LARP1 binding coverage across the 5'UTR, CDS, and 3'UTR regions of its targets, which were separated into non-translation-related (non-TR) genes, translation-related (TR) genes, and the subset of TR genes encoding RPs (*Figure 1B*; *Supplementary file 3*). Strikingly, LARP1 binding at 5'UTRs of TR genes more than doubled upon mTOR-inactivation, and binding on RP genes increased from 0% to 17% (*Figure 1B*). To further explore this observation, we plotted the accumulation of LARP1 binding under growing and mTOR-inactive conditions along normalized gene lengths (*Figure 1C*). Across non-TR mRNAs, LARP1 preferentially associated with CDSs and 3'UTRs, but was almost completely absent from 5'UTRs. In contrast, LARP1 bound most strongly to 3'UTRs of TR and RP mRNAs under growing conditions. Importantly, under conditions of mTOR inactivation, LARP1 accumulated at 5'UTRs, with the majority of 5'UTR binding occurring on RP transcripts (*Figure 1C*).

Since LARP1 regulates PES-containing mRNAs, including 5'TOP sequences, we searched the 58 and 92 5'UTR LARP1 binding sites under growing and mTOR-inactive conditions, respectively, for a consensus motif using MEME. We identified six consecutive pyrimidines in all 5'UTR LARP1 binding sites, suggesting that LARP1 binds directly to PESs (*Figure 1D*). Surprisingly, 5'UTR LARP1 binding sites rarely overlapped with 5'TOP sequences, which are located at the 5'-most end of 5'UTRs; instead, LARP1 binds predominantly at the 3'-most end of 5'UTRs (*Figure 1C and H*; *Supplementary file 4*). In fact, 5'TOP-containing 5'UTRs bound by LARP1 are more pyrimidine-rich at their 3' ends than those not bound by LARP1 (*Figure 1E*). To confirm that LARP1 binds PESs within target 5'UTRs, we compared pyrimidine-richness of LARP1-bound regions to non-LARP1-bound regions of these 5'UTRs and observed a significantly higher proportion of pyrimidines in LARP1-bound regions under both growing (Welch's two-tailed *t*-test: p=1.4e-15) and mTOR-inactive (p=1.2e-18) conditions (*Figure 1F*). Taken together, our data suggest that LARP1 specifically recognizes and binds PESs at the 3'-end of 5'UTRs for a subset of TR and RP transcripts and that LARP1 is not a genuine 5'TOP RNA binding protein (RBP) in vivo.

LARP1 also binds CDSs of non-TR genes and 3'UTRs of TR and RP mRNAs. We identified GA-rich motifs in 9–15% of these sites under both conditions (*Figure 1—figure supplement 1C*). LARP1-bound 3'UTR regions are slightly, but significantly, enriched for higher G-content than non-LARP1-bound regions on the same 3'UTRs (*Figure 1—figure supplement 1D*). These motifs are similar to ones identified for RRM domain-containing RBPs in a recent systematic in vitro study characterizing the sequence-specific recognition sites for RBPs across 24 eukaryotes (*Ray et al., 2013*), suggesting a possible role of LARP1's RRM domain (*Bayfield et al., 2010*) to interact with CDSs and 3'UTRs.

The relationship between LARP1 binding and decreases in translational efficiency (TE) upon mTOR inactivation are paralleled in mouse embryonic fibroblasts. We obtained measurements of changes in mouse transcript TE upon treatment of cells with Torin1 (*Thoreen et al., 2012*). Thirty-three percent of human homologs of mouse genes exhibiting decreased TE were bound by LARP1 in mTOR-inactive conditions compared to only 12% and 14% of genes showing no change in or increased TE, respectively (*Figure 1—figure supplement 1E*). We next wondered if increased pyrimidine richness observed at 3' ends of LARP1-bound 5'UTRs is functionally linked to mTOR-dependent changes in TE rates. We compared 5'UTR pyrimidine content of mouse RP-encoding mRNAs exhibiting the greatest changes in TE to those exhibiting the least and saw no difference at the 5'-most region of the 5'UTRs (*Figure 1—figure supplement 1F*). Strikingly, however, the 3'-most

5'UTR region of the most affected genes contained a significantly higher proportion of pyrimidines compared to the least affected genes (Welch's two-tailed *t*-test: p=0.036), indicating that pyrimidine richness at LARP1-interacting regions of 5'UTRs is correlated with strong decreases in TE upon mTOR inactivation. Together, these results suggest that the relationship between LARP1 binding and decreased TE is conserved from mouse to human.

Of the 88 annotated human ribosomal proteins, mRNAs encoding 84 were expressed in our mRNA-seq libraries under both conditions. Summing LARP1 binding site coverage of these genes confirms increased LARP1 binding at 5'UTRs and slightly decreased LARP1 binding at CDS and 3'UTR upon mTOR inactivation (*Figure 1G*). We verified the specific interaction between endogenous LARP1 and mRNAs encoding RpS6, S3A, S18, L26, and L32 by RNA immunoprecipitation (RIP) assays followed by quantitative PCR (qPCR) (*Figure 1—figure supplement 1G*). Taken together, these observations raise the intriguing possibility that the function of LARP1 in regulating RP mRNA translation may be context-dependent: the interaction of LARP1 with PESs in the 5'UTRs of RP mRNAs may have an inhibitory role, whereas its interaction with 3'UTRs may exert a positive role in RP mRNA translation.

## LARP1 is a direct substrate of Akt/S6K1 and mTORC1

To investigate the mechanisms by which site-specific LARP1 interaction with RP mRNAs is regulated by the activity of mTOR, we examined the roles of post-translational modifications of LARP1 regulated by mTOR activity. Previous genome-wide phospho-mass spectrometry analyses showed that LARP1 is a highly phosphorylated protein and has Torin1-sensitive phosphorylation sites (*Hsu et al., 2011*; *Kang et al., 2013*; *Yu et al., 2011*), suggesting that phosphorylation of LARP1 may regulate the configurations of LARP1-RP mRNA interaction. Phospho-mass spectrometry analysis targeting endogenous LARP1 revealed that more than 10 LARP1 phosphorylation sites, eight (highlighted by red color) of which were significantly sensitive to short-term Torin1 treatment (*Figure 2A* and *Figure 2—figure supplement 1A*), were identified. Among the Torin1-sensitive phosphorylation sites, two serine residues (Ser770 and Ser979), which follow the typical consensus motif (RxRxx[S/T]) for AGC kinases (*Figure 2B*), were directly phosphorylated by S6K1 or Akt in vitro (*Figure 2C and D*, and *Figure 2—figure supplement 1B*). Further, in vitro kinase assays using multiple polypeptides containing these Torin1-sensitive phosphorylation sites identified that Ser689 and Thr692, which do not match the AGC kinase phosphorylation motif (*Figure 2B*), were directly phosphorylated by mTOR (*Figure 2E*). These observations indicate that LARP1 is a direct substrate of mTOR, S6K, and Akt. Overexpression of active S6K1 or Akt induced phosphorylation of the RxRxx[S] motifs of wild type LARP1 but not the LARP1 2A mutant, where both Ser770 and Ser979 were substituted with alanine, in vivo (*Figure 2F and G*). Active Akt-induced LARP1 phosphorylation was not inhibited by rapamycin or S6K1 inhibitor (PF-4708671: PF), confirming that Akt is also able to directly phosphorylate LARP1 in vivo (*Figure 2—figure supplement 1C*). The phosphorylation of AGC kinase sites of LARP1 is sensitive to amino acids or amino acids/growth factor stimulation (*Figure 2H*). To investigate the functional roles of endogenous S6K1 and Akt in the phosphorylation of LARP1, we examined the phosphorylation status of LARP1 under acutely stimulated (growth factor/amino acid starvation for 60 min then stimulation with growth factor/amino acid for 10 min) or steady state growth conditions with or without the addition of rapamycin, Torin1, or a specific Akt inhibitor, MK-2206 (*Figure 2I*). Upon growth factor/amino acid stimulation, the phosphorylation of both Akt and S6K1 was enhanced compared to those under starvation conditions. Simultaneously, the phosphorylation of LARP1, as detected by both the Akt substrate antibody and the pS979 LARP1 antibody, was enhanced. Importantly, under this acutely stimulated condition, rapamycin, which completely inhibits S6K1, but not Akt, had little effect on S770/S979 phosphorylation of LARP1. In contrast, the Akt inhibitor MK-2206, as well as Torin1, which inhibit both Akt and S6K1, largely inhibited S770/S979 phosphorylation of LARP1.

Under the steady state growth condition, inhibition of S6K1 with rapamycin or PF 4708671, a S6K inhibitor, equally and significantly decreased LARP1 phosphorylation, and Torin1 and MK-2206 further blocked these phosphorylations. Taken together, these observations indicate that Akt is a physiologically relevant primary kinase for S770/S979 phosphorylation of LARP1 especially under acute stimulatory conditions. Under steady state conditions, S6K plays a major role while Akt also partially contributes to the phosphorylation of LARP1 (*Figure 2J*).

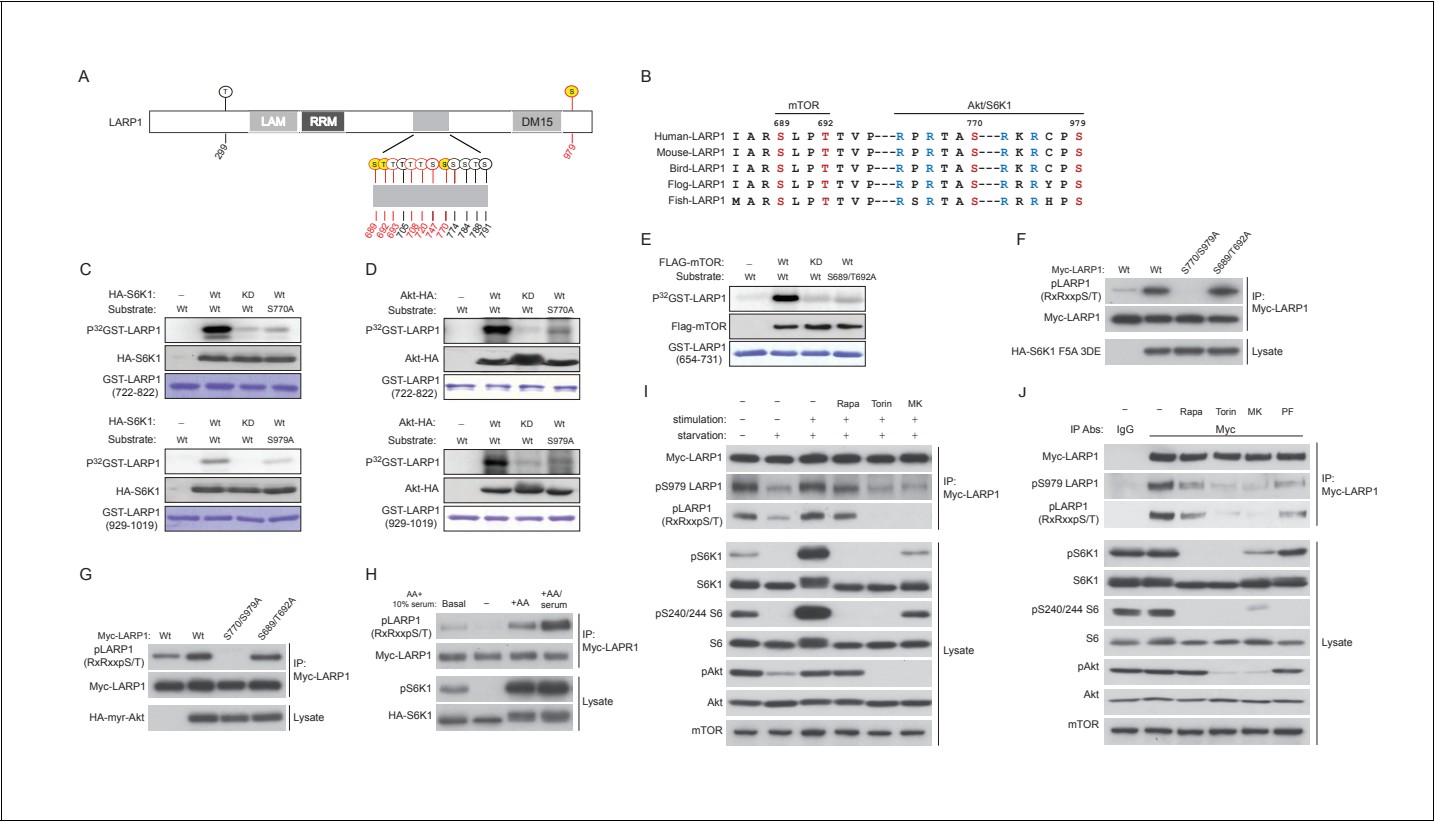

**Figure 2.** LARP1 is a direct substrate of mTOR, Akt, and S6K1. (**A**) Schematic position of LARP1 phosphorylation sites identified by liquid chromatography coupled to electrospray ionization tandem mass spectrometry (LC-ESI-MS/MS). (**B**) Location and sequence conservation of LARP1 phosphorylation sites. (**C–E**) S6K1 (**C**), Akt (**D**), and mTOR (**E**) directly phosphorylate LARP1 in vitro. In vitro kinase assay (IVK) were performed with the indicated wild type kinase (WT) and inactive kinase (KD) purified from HEK293T cells using the indicated GST-LARP1 fragments. (**F–G**) Active S6K1 (**F**) or Akt (**G**) enhances phosphorylation of wild-type LARP1 but not the S770A/S979A LARP1 mutant in HEK293T cells. Phosphorylation of LARP1 was detected by phospho-specific-Akt substrate antibody. (**H**) Levels of LARP1 phosphorylation sites of AGC kinases are enhanced by amino acids or amino acids/growth factors. (**I**) Amino acids/growth factors-inducible S770/S979 phosphorylation of LARP1 is partially inhibited by rapamycin but largely inhibited by Torin1 or MK-2206. HEK293T cells were serum starved over night and incubated with HBSS with or without the indicated inhibitors for 1 hr before stimulation with DMEM containing 10% FBS for 10 min. (**J**) Levels of S770/S979 phosphorylation of LARP1 are decreased by rapamycin or S6K1 inhibitor (PF 470861) and further decreased by Torin1 or MK-2206 under steady state growth conditions.

The following figure supplement is available for figure 2:

**Figure supplement 1.** LARP1 is a direct substrate of mTOR, Akt, and S6K1.

## LARP1 functions as a phosphorylation-sensitive molecular switch for RP mRNA translation

Our PAR-CLIP data indicated that while LARP1 primarily interacts with 3'UTRs of RP mRNAs under growth conditions, it also binds to 5'UTRs of RP mRNAs upon mTOR inactivation (*Figure 1C*). To investigate the roles of LARP1 phosphorylation in binding RP mRNAs, we examined specific interactions between LARP1 and the 5' or 3'UTR of RpL32 mRNA as a representative example of RP mRNAs bound by LARP1 identified in our PAR-CLIP analyses (*Figure 1H*). In agreement with our PAR-CLIP data, RIP assays demonstrated that endogenous LARP1 interacted with both the 5'UTR (closed bar) and 3'UTR (open bar) of RpL32 under amino acid starvation conditions (*Figure 3A*). In addition, in response to amino acids, which enhances mTORC1 activity, the interaction between LARP1 and the RpL32 5'UTR was decreased in a manner dependent on cellular mTOR kinase activity. In contrast, the binding of LARP1 to the RpL32 3'UTR was increased in response to mTOR activation.

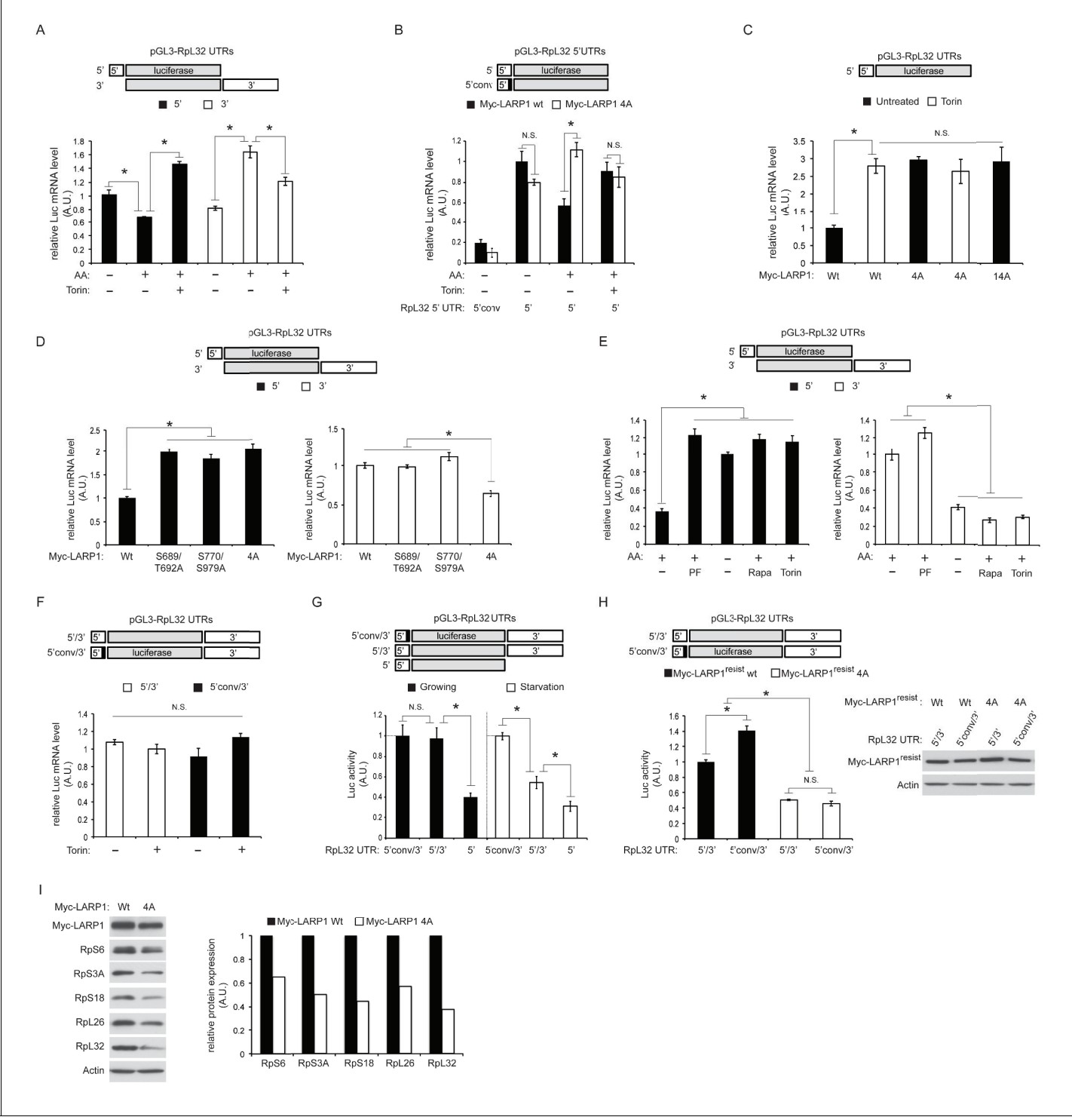

**Figure 3.** Dynamic rearrangement of LARP1 biding to the UTRs of RpL32 mRNA is regulated by the phosphorylation of LARP1. (**A**) The effect of amino acid and mTOR inhibitor on the levels of endogenous LARP1 binding to the 5' and 3'UTR of RpL32 mRNA. HEK293T cells were transfected with the indicated reporter mRNAs. Endogenous LARP1 was IPed, and the levels of co-IPed luciferase mRNA were determined by qPCR. Data were normalized by input luciferase mRNAs and the amount of IPed LARP1. *p<0.05, mean±SEM (n = 3). (**B**) LARP1 phosphorylation by mTOR and S6K1/Akt induces its dissociation from the PES in the 5'UTR of RpL32 mRNA. The wild type and the LARP1 4A mutant were transfected with the indicated reporter mRNAs. Data were expressed as *Figure 3A*. N.S. denotes 'not significant'. *p<0.05, mean±SEM (n = 3). (**C**) The effect of alanine substitutions of all the phosphorylation sites of LARP1 identified in this study on the binding to the 5'UTR of RpL32 mRNA under growth conditions. N.S. denotes 'not significant'. *p<0.05, mean±SEM (n = 3). (**D–E**) Both mTOR- and S6K1/Akt-dependent LARP1 phosphorylation are necessary for its dissociation from the

*Figure 3 continued on next page*

*Figure 3 continued*

5'UTR of RpL32 mRNA, while either mTOR or S6K1/Akt phosphorylation of LARP1 is sufficient to maintain its binding to the 3'UTR. The wild type and the indicated LARP1 mutants were IPed, and the levels of co-IPed 5' or 3' reporter mRNA were determined by qPCR (D). HEK293T cells were starved with amino acids or treated with the indicated inhibitors for 1 hr, and levels of co-IPed 5' or 3' reporter mRNA with endogenous LARP1 were determined (E). Data were normalized by input luciferase mRNAs and the amount of IPed LARP1. *p<0.05, mean±SEM (n = 3). (F) LARP1 constitutively interacts with the RpL32 reporter RNA containing both the 5' and 3' UTRs in a manner independent of the PES in the 5'UTR and mTOR activity. Endogenous LARP1 PAR-CLIP was performed in the presence or absence of Torin1 treatment. Levels of LARP1-bound reporter mRNA were determined by qPCR. Data were normalized by input luciferase mRNAs and the amount of IPed LARP1. N.S. denotes 'not significant', mean±SEM (n = 3). (G) The PES motif on 5'UTR of RpL32 is the cis-acting element necessary for translation inhibition in response to mTORC1 inhibition. HEK293T cells were transfected with the indicated reporter mRNAs and grown in normal growth media (10% serum) or mild starvation media (1% serum). After 24 hr, luciferase activity was measured and normalized by luciferase mRNA levels. *p<0.05, mean±SEM (n = 3). (H) Phosphorylation of LARP1 and its dissociation from the 5'UTR are critical for the translation of RpL32. HEK293T cells lacking endogenous LARP1 were transfected with the indicated shRNA-resistant LARPs and reporter mRNAs. 48 hr post-transfection, luciferase activity was measured and normalized by luciferase mRNA levels (left panel). *p<0.05, mean±SEM (n = 3). Levels of transfected Myc-LARP1s were shown (right panel). (I) Phosphorylated LARP1 plays a positive role in ribosomal protein translation. Myc-tagged wild type LARP1 and the 4A mutant LARP1 were stably expressed in HEK293T cells by retrovirus-mediated infection to achieve lower levels of LARP1 expression, and endogenous LARP1 was knockdown by LARP1 shRNA targeting its 5'UTR. Levels of RP proteins were determined by western blotting and the intensity of the bands was quantified.

Importantly, phosphorylation of LARP1 (S689/T692 and S770/S979) identified in *Figure 2* plays key roles in LARP1 binding to both the 5' and 3' UTRs of RpL32 mRNA. Similar to endogenous LARP1, the interaction of wild-type myc-LARP1 (closed bar) with the 5'UTR of RpL32 is decreased by amino acid stimulation in a manner dependent on mTOR activity (*Figure 3B*). However, alanine substitutions of mTOR (S689/T692) and S6K1/Akt (S770/S979) phosphorylation sites confer the LARP1 4A mutant (open bar) resistant to its dissociation from the 5'UTR in response to amino acid stimulation. Furthermore, the PES located at the 3' end of the RpL32 5'UTR was essential for its interaction with LARP1 as substitutions of pyrimidines with guanines (5'conv) in the PES region within the 5'UTR (*Figure 1H*) dramatically reduced its interaction with both wild type LARP1 and LARP1 4A mutant even under amino acid starvation conditions (*Figure 3B*). The LARP1 14A mutant, which bears an additional 10 alanine substitutions in the phosphorylation sites identified in *Figure 2A*, also displayed enhanced binding to the 5'UTR of RpL32 mRNA, similar to the LARP1 4A mutant compared to wild type LARP1 under growth culture conditions (*Figure 3C*). These results indicate that phosphorylation of these four serine/threonine residues plays a key role in reducing the interaction of LARP1 with the 5'UTR of RpL32 mRNA through the PES. Further analyses revealed that both mTOR- and S6K1-dependent LARP1 phosphorylation were required to reduce its binding to the 5'UTR of RpL32 mRNA, while either mTOR- or S6K1-dependent phosphorylation of LARP1 was sufficient to maintain its association with the 3'UTR under steady state growth condition (*Figure 3D*). Consistently, the inhibition of S6K1 activity with PF-4709671 sufficiently blocked the reduction of LARP1's binding to the 5' UTR of RpL32 mRNA as did amino acid starvation, rapamycin or Torin1 treatment, which inhibits both S6K1 and mTORC1 activity. However, PF-470961 treatment alone failed to induce the dissociation of LARP1 from the 3'UTR of RpL32 mRNA (*Figure 3E*). Together with our PAR-CLIP data, these observations indicate that the affinity of LARP1 binding to the 5' and 3' UTR of RpL32 mRNA dynamically changes in a manner dependent on mTOR and S6K1/Akt-dependent phosphorylation.

To examine the configuration of LARP1 binding to the 5'UTR and 3'UTR of RpL32 mRNA, we performed PAR-CLIP assays using endogenous LARP1 with the reporter mRNAs containing both the 5' and 3'UTR of RpL32 mRNA. Interestingly, under cross-linking conditions, LARP1 can bind the wild type (5'/3') or the 5'UTR-mutated (5'conv/3') reporter mRNA equally well in the presence or absence of Torin1 treatment, respectively (*Figure 3F*). These observations suggest that the same amount of LARP1 constantly interacts with the UTRs of RpL32 mRNA under both high and low mTOR activity conditions, although the affinity of LARP1 binding to the 5' and 3' UTR of RpL32 mRNA significantly changes in response to cellular mTOR/Akt activity. This raises the possibility that non-phosphorylated LARP1 may simultaneously interact with both the 5' and 3' UTR of the same RpL32 mRNA under low mTOR activity conditions while phosphorylated LARP1 dissociates from the 5'UTR and mainly interacts with the 3'UTR of RpL32 mRNA.

To examine the significance of LARP1 interaction with the 5' or 3'UTR in the translation of RpL32 mRNA, we measured luciferase production from the indicated reporters (*Figure 3G*). Luciferase protein expression from the reporter containing both 5' and 3'UTRs of RpL32 was consistently higher compared to the reporter with just 5'UTR, suggesting that the 3' UTR is important for stimulating RpL32 translation. Both the 5' UTR-mutated reporter (5'conv/3') and the wild type reporter (5'/3') generate similar levels of luciferase protein under growth conditions (*Figure 3G*, filled bars) that induce LARP1 dissociation from the 5'UTR. Importantly, under mild serum starvation conditions, the 5'UTR-mutated reporter (5'conv/3') produces more luciferase protein compared to the wild type reporter (5'/3') (*Figure 3G*, open bars). Furthermore, phosphorylatable wild type Myc-LARP1 showed some advantages in stimulating translation of the 5'conv/3' reporter compared to the wild type 5'/3' reporter in cells lacking endogenous LARP1 (*Figure 3H*). In contrast, the phopho-defective Myc-LARP1 4A mutant significantly and equally inhibited the translation of both reporters, regardless of the existence (5'/3') or absence (5'conv/3') of the LARP1 binding site in the 5'UTR in these cells. These observations indicate that the dissociation of LARP1 from the PES motif in the 5'UTR and the phosphorylation of LARP1, which enhances LARP1 binding to the 3'UTR, are important processes in stimulating RpL32 mRNA translation. Moreover, the data also suggest that the phosphorylation of LARP1 has an additional key role in stimulating the translation of RpL32 mRNA at the 3'UTR (see Figures 5 and 6). Note that the translation of the 5'conv/3' mutant reporter was slightly, but significantly, enhanced compared to the 5'/3' wild type reporter in cells expressing ectopic Myc-LARP1, which was not seen in the cells expressing just endogenous LARP1 (*Figure 3G*). We speculate that overexpressed Myc-LARP1 may not be fully phosphorylated in our experimental conditions. As a consequence, non-phosphorylated wild type LARP1 may still bind to the PES in the 5'UTR and reduce the translation of the wild type 5'/3' reporter. Accordingly, cells expressing the LARP1 4A mutant, which strongly binds to the 5'UTR of RpL32 mRNA irrespective of cellular mTOR activity, have reduced expression of RP proteins including RpL32 (*Figure 3I*). Together with our PAR-CLIP data (*Figure 1*), these observations suggest that LARP1 stimulates RP mRNA translation through its interaction with the 3'UTR and that its association with the 5'UTR has a negative role in RP mRNA translation.

## LARP1 recruits mTORC1 to LARP1-interacting mRNPs in a manner dependent of mTORC1 activity

To stimulate the translation of 5'TOP mRNAs, previous studies have proposed that LARP1 directly interacts with initiation factors and polyA-binding protein (PABP) in response to mTORC1 activity (*Aoki et al., 2013*; *Tcherkezian et al., 2014*). In contrast, a more recent study proposed that active mTORC1 interacts with LARP1 and inhibits LARP1's function to suppress translation initiation of LARP1-interacting 5'TOP mRNAs (*Fonseca et al., 2015*). Thus, the functional importance of LARP1-mTORC1 interaction and the role of LARP1 in the regulation of translation of LARP1-interacting mRNAs remain unclear. We also observed that endogenous as well as exogenous LARP1 specifically and stably interacts with mTORC1 through Raptor under growth conditions (*Figure 4A* and *Figure 4—figure supplement 1A–C*). LARP1 exclusively interacts with mTORC1 but not with mTORC2 (*Figure 4A* and *Figure 4—figure supplement 1A*). mTOR sufficiently co-IPs LARP1 only in the presence of endogenous Raptor (*Figure 4—figure supplement 1B*). In addition, Raptor is able to co-IP LARP1 in the presence of non-ionic detergent (NP-40), which is known to disrupt the interaction between Raptor and mTOR (*Figure 4—figure supplement 1C*). These results indicate that LARP1 association with mTORC1 requires Raptor.

LARP1 also co-IPs PABP1, a polyA tail and eIF4G1 binding protein, as previously reported (*Blagden et al., 2009*; *Burrows et al., 2010*; *Tcherkezian et al., 2014*). However, whether LARP1 directly interacts with PABP1 or indirectly, perhaps through a common mRNA substrate, remains unresolved. To address the nature of LARP1 interaction with PABP1 and mTORC1, we treated cell lysates with RNaseA prior to or after LARP1 immunoprecipitation (*Figure 4B* and *Figure 4—figure supplement 1D*). As expected, LARP1 co-IPed not only mTORC1 and PABP1, but also eIF4E, a 5'CAP mRNA binding initiation factor. While LARP1 interaction with mTORC1 was resistant to RNaseA treatment, its interaction with PABP1 and eIF4E was markedly sensitive to RNaseA treatment. (*Figure 4B* and *Figure 4—figure supplement 1D*). In addition, eIF4G1, a scaffolding protein for forming the eIF4F complex, co-IPed other eIF4F components and LARP1 (*Figure 4B*). Although the interaction between eIF4G1 and other eIF4F components such as eIF4E, eIF3B, and PABP1 was

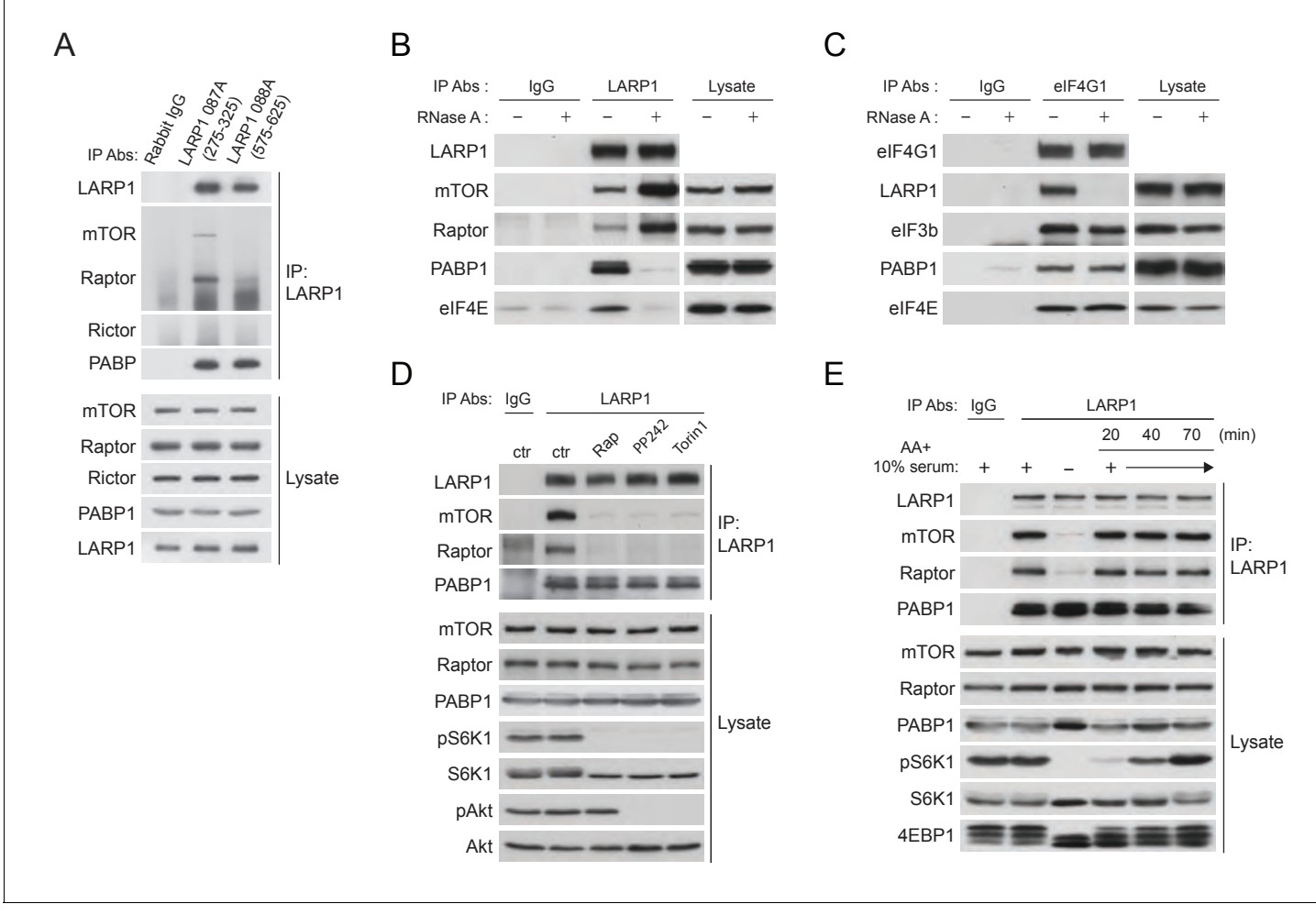

**Figure 4.** LARP1 recruits mTORC1 to LARP1-interacting mRNPs in a manner dependent of mTORC1 activity. (**A**) Endogenous LARP1 co-IPs endogenous mTORC1 in HEK293T cells. LARP1 antibody 087A, but not 088A, co-IPs mTORC1. The LARP1 antibody, 087A or 088A recognizes amino acids 275–325 or 575–625 of LARP1, respectively. (**B**) The effect of RNAse A on the interaction of LARP1 with mTORC1 and other mRNA binding proteins (PABP1 and eIF4E). Endogenous LARP1 was IPed from the lysates treated with RNAse A. (**C**) The effect of RNase A on the interaction of LARP1 with the components of the initiation complex. Endogenous eIF4G1 was IPed from the lysates treated with RNAse A. (**D**) LARP1 interacts with mTORC1 in a manner dependent on mTORC1 activity. (**E**) LARP1 co-IPs mTORC1 in a growth factor/amino acid stimulation-dependent manner.

The following figure supplement is available for figure 4:

**Figure supplement 1.** LARP1 interacts with mTORC1 in an RNAse A insensitive manner.

RNaseA resistant, the binding of eIF4G1 to LARP1 was abolished by the treatment with RNaseA. Interestingly, LARP1 coIPed more mTORC1 from lysates treated with RNaseA than in untreated lysates (*Figure 4B* and *Figure 4—figure supplement 1D*), suggesting that a substantial pool of the mTORC1-LARP1 complex is refractory to IP with the LARP1 antibody under standard lysis conditions and may exist in an RNaseA-sensitive pool. Taken together, these data indicate that LARP1 indirectly associates with PABP1 and initiation factors through binding of common mRNAs, while LARP1-mTORC1 interaction occurs through direct protein-protein contacts.

Importantly, the activity of mTORC1 plays a key role in LARP1's binding to mTORC1 but not to PABP1. LARP1 failed to interact with mTORC1 but maintained its interaction with PABP1 subjected to mTOR inhibitors, suggesting that LARP1 associates with mRNAs regardless of cellular mTOR activity (*Figure 4D*), consistent with our PAR-CLIP data (*Figure 1C*). Furthermore, the interaction between endogenous LARP1 and mTORC1 was quickly and fully restored by replenishment of amino

acids and growth factors after starvation of these activating cues, whereas the interaction between LARP1 and PABP1 was not affected by these cues (*Figure 4E*). Furthermore, recovery of the interaction between LARP1 and mTORC1 by growth factor/nutrient stimulation occurred concomitantly with 4EBP1 phosphorylation but precedes full phosphorylation of S6K1.

## LARP1 scaffolds mTORC1 to LARP1-interacting mRNAs in a manner dependent on LARP1 phosphorylation

To investigate the mechanisms underlying the formation of the LARP1-mTORC1 complex, we determined regions of LARP1 necessary to associate with mTORC1. Interestingly, serial truncations of the LARP1 carboxyl terminus revealed that the DM15 motif and an N-terminal region adjacent to the DM15 motif (*Figure 5A*), where the majority of Torin1-sensitive LARP1 phosphorylations occur (*Figure 2A*), were critical for its association with mTORC1. The LARP family of proteins consists of

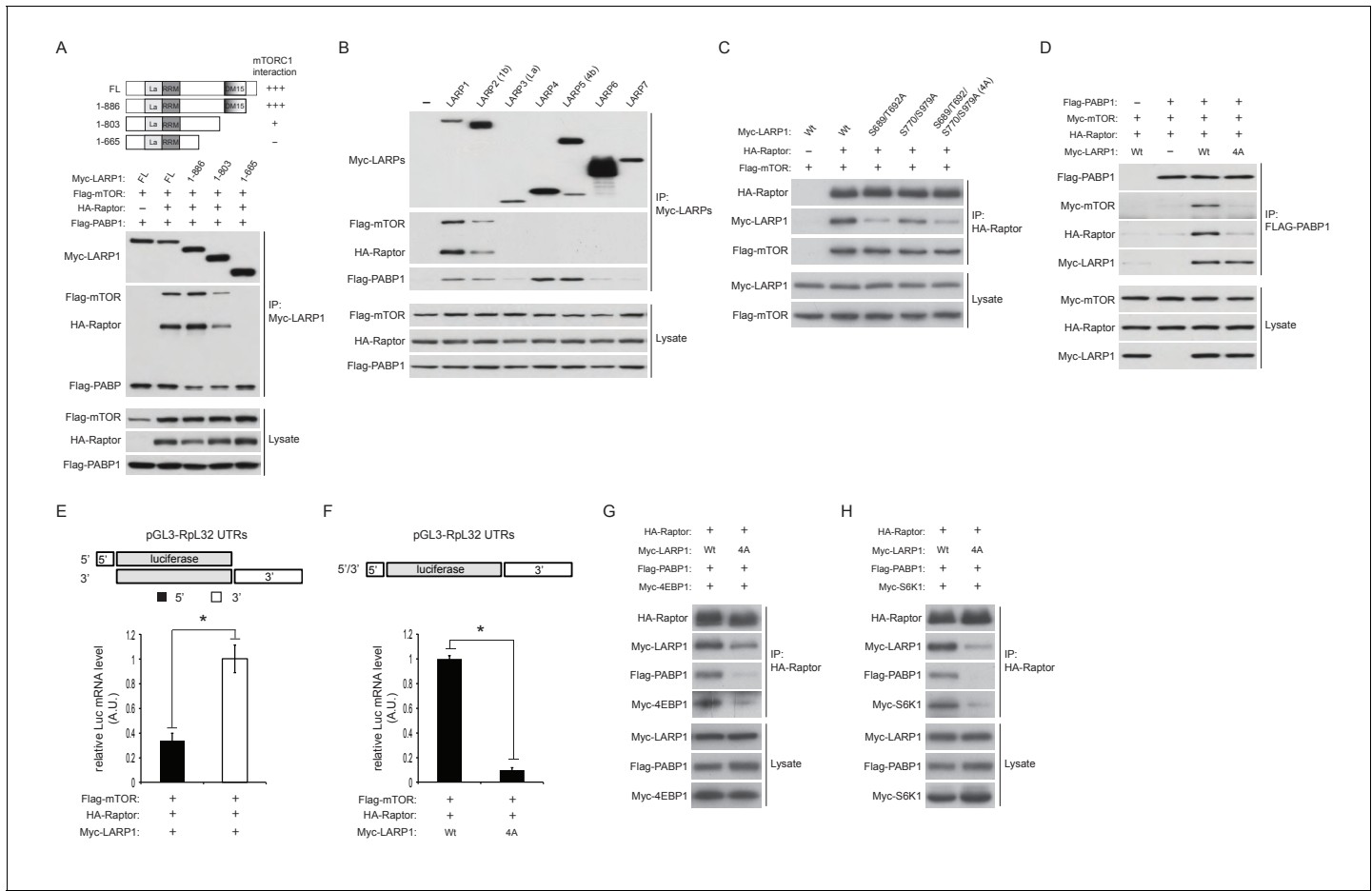

**Figure 5.** LARP1 scaffolds mTORC1 to LARP1-interacting mRNAs in a manner dependent on LARP1 phosphorylation. (**A**) The DM15 domain and its adjacent N-terminal region of LARP1 are required for the interaction with mTORC1. (**B**) LARP1 and LARP2 but not other LARP family members interact with mTORC1. (**C**) mTORC1-dependent LARP1 phosphorylation (S689/T692) plays a major role in the interaction between phosphorylated LARP1 and mTORC1. (**D**) mRNPs containing wild type LARP1 (Wt), but not the phospho-defective LARP1 (4A), associate with mTORC1. (**E**) mTORC1 preferentially interacts with the 3'UTR than the 5'UTR of RpL32 mRNA. HA-Raptor RIP assays were performed in the presence of wild type LARP1 with the indicated reporter mRNAs. Data were expressed as *Figure 3A*. (**F**) mTORC1 interacts with the RpL32 reporter mRNA in the presence of wild type but not LARP1 4A mutant. HA-Raptor RIP assays were performed. (**G–H**) mTORC1 more interacts with its substrates, 4EBP1 (**G**) and S6K1 (**H**) in the presence of wild type LARP1 compared to the LARP1 4A mutant.

The following figure supplement is available for figure 5:

**Figure supplement 1.** Schematic structure of human LARP family.

six members comprised of LARP1, 2 (1B), 4, 5 (4B), 6, and 7 (*Figure 5—figure supplement 1*) (*Bousquet-Antonelli and Deragon, 2009*). Consistently, only LARP1 and LARP2, both of which possess the DM15 motif and an adjacent N-terminal region, interacted with mTORC1 (*Figure 5B*). Although LARP2, a close LARP1 paralog (59% amino acid sequence identity), interacted with mTORC1, its affinity for mTORC1 was significantly weaker than LARP1. These data suggest that the phosphorylations of LARP1 may also play an important role in maintaining LARP1-mTORC1 interactions, in addition to their roles in the binding of LARP1 with RP mRNAs.

To examine the role of LARP1 phosphorylation in the regulation of interaction between LARP1 and mTORC1, we first assessed mTORC1 binding with a series of phospho-defective LARP1 mutants (S689/T692A: mTOR sites, S770/S979A: Akt/S6K1 sites, and 4A: mTOR and Akt/S6K1 sites). While the replacement of LARP1 Akt/S6K1 phosphorylation sites with alanines only slightly reduced LARP1 interaction with mTORC1, mutations in the LARP1 mTOR phosphorylation sites largely abrogated its interaction with mTORC1 (*Figure 5C*). Importantly, both the wild type and LARP1 4A mutant were able to associate with PABP1 equally well through their binding to common mRNAs; however, PABP1 co-IPed mTORC1 only in the presence of wild-type LARP1 but not the LARP1 4A mutant (*Figure 5D*). These observations suggest that LARP1 phosphorylation by mTORC1 is critical for tethering mTORC1 to the LARP1-mRNP complex. In support of this hypothesis, under growth conditions where phosphorylated LARP1 mainly binds to the 3′UTRs of RP mRNAs (*Figures 1B, C* and *3A*), mTORC1 associated more with the 3′UTR than the 5′UTR of RpL32 mRNA in Raptor CLIP assays (*Figure 5E*). mTORC1 association with RpL32 mRNA was dependent on wild type LARP1 but not the LARP1 4A mutant (*Figure 5F*). Moreover, wild type LARP1 supports greater association of mTORC1 with its substrates S6K1 and 4EBP1 than the LARP1 4A mutant (*Figure 5G and H*). Taken together, these data suggest that while non-phosphorylated LARP1 blocks RP mRNA translation through its interaction with the 5′UTR of RP mRNAs, phosphorylated LARP1 is converted into a scaffolding protein for mTORC1 on the 3′UTRs of RP mRNAs to facilitate the accessibility of mTORC1 with 4EBP1 and S6K1.

To further investigate the roles of LARP1 in mTORC1-dependet phosphorylation of its substrates, endogenous LARP1 was knocked down in mammalian cells with varying degrees of insulin sensitivity. In HEK293T cells (insulin insensitive), ablation of LARP1 slightly decreased phosphorylation of S6K1, S6, and 4EBP1 compared to those in control cells under normal growth culture conditions (*Figure 6—figure supplement 1A*). However, in response to growth factor/amino acids stimulation, LARP1 knockdown significantly delayed and attenuated phosphorylation of these proteins in HEK293T cells (*Figure 6A* and *Figure 6—figure supplement 1A*), LnCap (prostate cancer cell line) (*Figure 6—figure supplement 1B*), and HEK293E cells (insulin sensitive) (*Figure 6—figure supplement 1C*). In contrast, Akt phosphorylation (Ser473) was not affected by the ablation of LARP1, indicating that the activity of PI3K and mTORC2 was intact in the LARP1 knockdown cells. In addition, LARP1 knockdown had little effect on intrinsic mTOR kinase activity monitored by mTOR auto-phosphorylation (Ser2481) (*Figure 6A*, *Figure 6—figure supplement 1A–C*) (*Copp et al., 2009*; *Peterson et al., 2000*), or on the integrity of mTORC1 and its kinase activity as measured by in vitro kinase assays (*Figure 6—figure supplement 1D*). The inhibitory effects of LARP1 knockdown on inducible S6K1 and 4EBP1 phosphorylation were not due to a disruption of eIF4F-mediated translation because knockdown of eIF4G1, which plays a key role in mTORC1-dependent translation (*Thoreen et al., 2012*), did not affect S6K1, S6, or 4EBP1 phosphorylation (*Figure 6B*).

Immunostaining of LARP1 and phosphorylated forms of 4EBP1 and S6 revealed that endogenous LARP1 was predominantly expressed in the cytoplasm where 4EBP1 and S6 phosphorylation occurs in response to growth factor and nutrient stimulation (*Figure 6C*). In addition, LARP1 did not colocalize with LAMP2 on the lysosomal membrane, where mTORC1 is activated. Consistent with the biochemical observations (*Figure 6A* and *Figure 6—figure supplement 1A–C*), LARP1 knockdown significantly reduced growth factor/nutrient-induced phosphorylation of 4EBP1 and S6 (*Figure 6C*). Notably, LARP1 knockdown (*Figure 6—figure supplement 1E*) or ectopic expression (*Figure 6D*) did not alter amino acid-induced mTORC1 localization on the lysosomal membrane, suggesting that, in addition to growth factor-induced PI3K/Akt activity, the amino acid-sensing mechanism on lysosomal membranes also remains intact in LARP1 knockdown and overexpressing cells. In response to amino acids, co-localization of Flag-LARP1 with mTOR became more obvious in cytosolic regions but not on the lysosomal membrane (*Figure 6D*, lower panels). Taken together, these data support

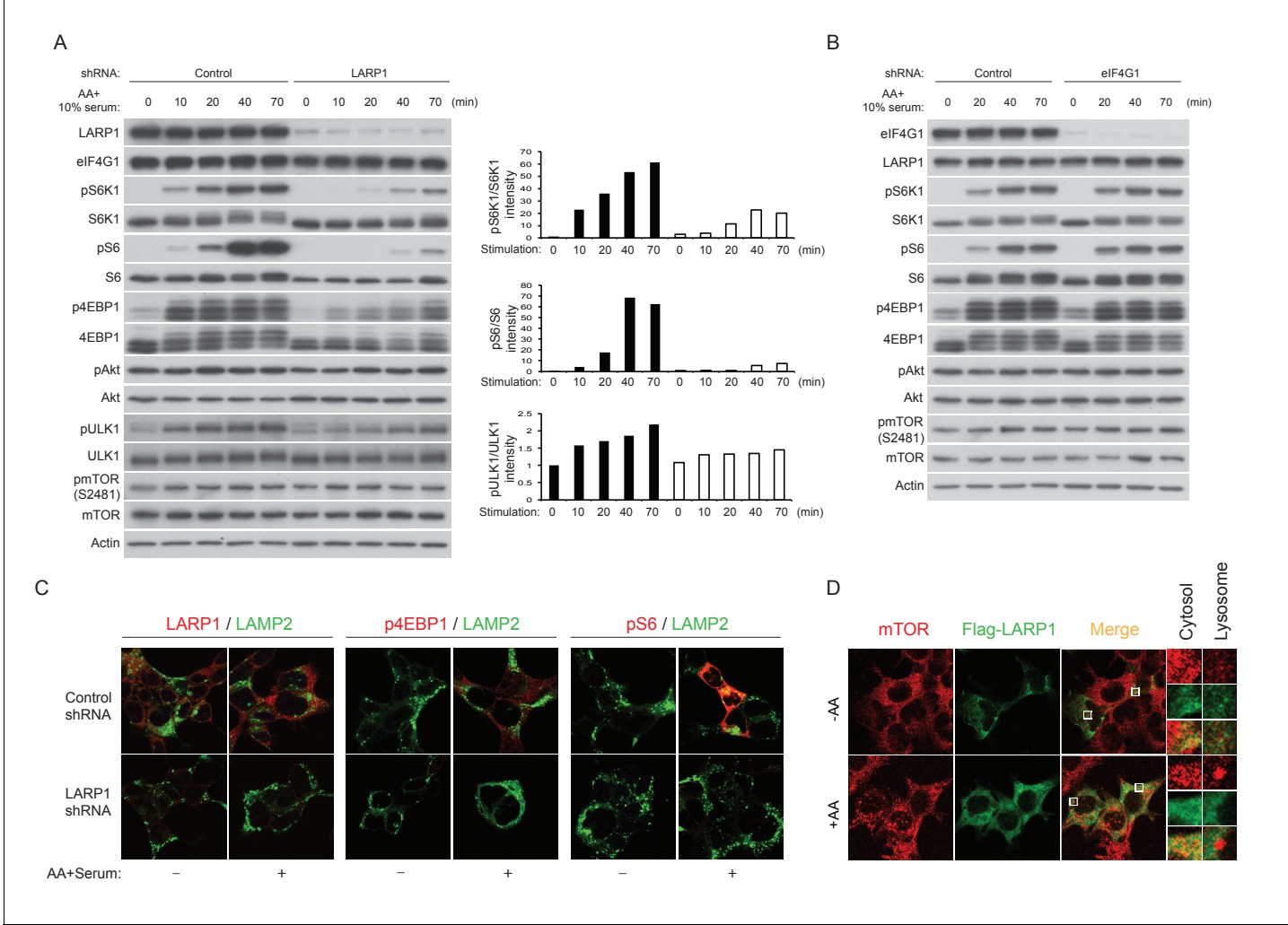

**Figure 6.** LARP1 enhances mTORC1-dependent phosphorylation of its substrates. (**A–B**) LARP1, but not eIF4G1, is required for growth factor/amino acid-induced phosphorylation of mTORC1 substrates in HEK293T cells. (**C**) LARP1 expresses in cytosolic region and supports growth factor/amino acid-induced S6 and 4EBP1 phosphorylation by mTORC1 in the cytosol. Note that neither LARP1 nor phosphorylated 4EBP1 co-localize with LAMP2, a lysosomal marker. (**D**) LARP1 co-localizes with mTOR at the cytosolic region in response to amino acid. Note that amino acid-inducible mTOR-positive punctate structures were shown as lysosomes.

The following figure supplement is available for figure 6:

**Figure supplement 1.** LARP1 enhances mTORC1-dependent phosphorylation of its substrates.

the idea that phosphorylated LARP1 facilitates mTORC1-dependent phosphorylation of S6K1 and 4EBP1 on the LARP1-containing mRNPs by scaffolding mTORC1.

## Loss of LARP1 function causes inefficient RP translation elongation

In agreement with the results of *Figure 6A*, m7GTP pull down assays showed that the interaction of 4EBP1 with eIF4E was increased in LARP1 knockdown cells (*Figure 7A*), suggestive of increased levels of non-phosphorylated 4EBP1 in LARP1 knockdown cells. Consistently, LARP1 knockdown partially decreased eIF4E in fractions containing pre-initiation complex (43S and 48S: fraction 5 and 6) while Torin 1, an mTOR kinase inhibitor, largely eliminated eIF4E from these fractions (*Figure 7B*). These data indicate that a fraction of cellular translation initiation is decreased in LARP1 knockdown cells. However, intriguingly, RP mRNAs that interact with LARP1 accumulated more in lighter polysome fractions in LARP1 knockdown cells compared to control cells under normal growth conditions

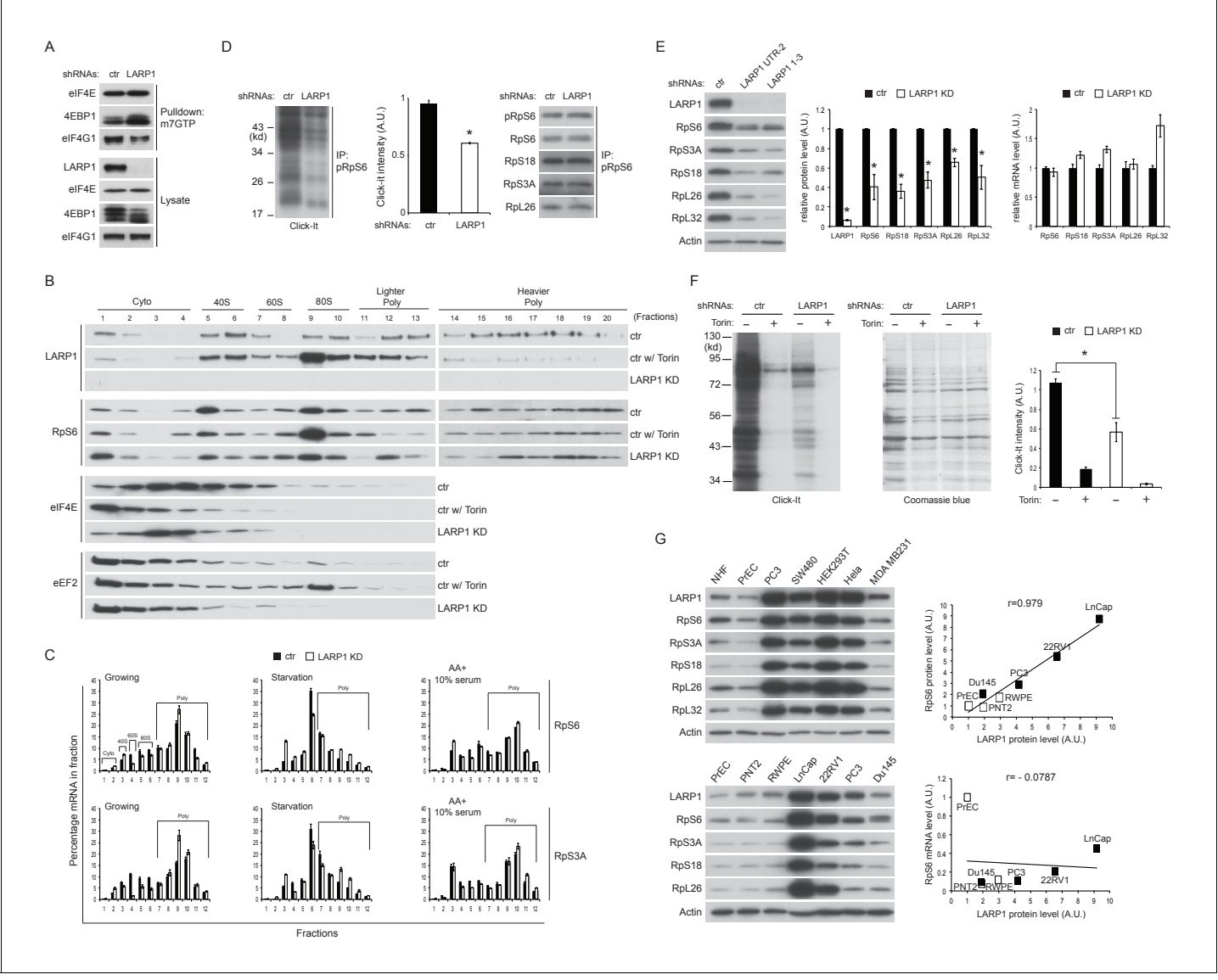

**Figure 7.** Loss of LARP1 causes defects in the multiple steps of RP mRNA translation. (**A**) LARP1 knockdown enhances 4EBP1 binding to the eIF4E precipitated with m7GTP sepharose beads. (**B**) Loss of LARP1 decreases the expression of eEF2 in the fractions containing active monosomes (80S) and polysomes. (**C**) RP mRNAs are accumulated in the lighter polysome fractions in LARP1 knockdown cells. (**D**) Loss of LARP1 decreases the translation of RP mRNAs. Equal amount of ribosomes were immunoprecipitated by phopho-S6 antibody from normal growing HEK293T cells in the presence or absence of LARP1 expression (right panel). Newly synthesized ribosome subunits were visualized (left panel) and quantified (middle panel). *p<0.05, mean±SEM (n = 3). (**E**) Prolonged LARP1 knockdown (96 hr) decreases the expression of RP proteins (left panels). Levels of RP proteins (middle panel) were quantified and mRNA levels of RP proteins were monitored by qPCR (right panel). Newly synthesized RP proteins were monitored by the Click-It assy *p<0.05 vs control shRNA treatment, mean±SD (n = 3). (**F**) Prolonged LARP1 knockdown (96 hr) decreases global protein synthesis. p<0.05, mean±SD (n = 3). (left panel). Equal amount of protein loading was visualized by Coomassie blue staining (middle panel). Click-It reaction was quantitated (right panel). *p<0.05 vs control shRNA treatment, mean±SD (n = 3). (**G**) The expression of LARP1 and RP proteins is enhanced in multiple cancer and transformed cell lines (left panels). Correlation between LARP1 protein vs. RpS6 protein (right upper) or RpS6 mRNA (right lower) in prostate epithelial cells. Open or close square indicates normal or benign prostate epithelial cells or metastatic prostate cancer cells, respectively.

The following figure supplements are available for figure 7:

**Figure supplement 1.** Loss of LARP1 causes defects in the multiple steps of RP mRNA translation.

**Figure supplement 2.** Hypothetical model for mTORC1-dependent LARP1-interacting RP mRNA translation.

(*Figure 7C*). In control cells, acute nutrient/growth factor starvation selectively redistributes RP mRNAs from the polysome fractions to the monosome (80S) fraction (*Figure 7C*, middle panels). In contrast, under the same starvation conditions, LARP1 knockdown results in retention of a substantial portion of RP mRNAs in the polysomes fractions while attenuating the accumulation of RP mRNAs in the monosome fraction (*Figure 7C*, middle panels). These observations suggest that RP mRNA translation is less sensitive to the inhibition caused by growth factor/nutrient depletion in LARP1 knockdown cells. Furthermore, replenishing growth factors and nutrients following identical starvation conditions showed similar RP mRNA distribution in polysome fractions in both control and LARP1 knockdown cells (*Figure 7C*, right panels). These results suggest that both ribosome dissociation from RP mRNAs and ribosome loading on RP mRNAs are inefficient, and that ribosome stalling may occur during RP mRNA elongation in LARP1 knockdown cells. However, non-LARP1 interacting mRNAs such as FOXO1 and YBX1 did not accumulate in polysome fractions in LARP1 knockdown cells, indicative of the specific roles of LARP1 in preventing ribosome stalling during the translation of LARP1 interacting mRNAs (*Figure 7—figure supplement 1A*). Indeed, in cells subjected to short-term LARP1 knockdown, the expression of eEF2 in the 80S monosome and polysome fractions was reduced (*Figure 7B*). Taken together, these data suggest that the efficiency of translation elongation of LARP1-interacting mRNAs is likely to be compromised in LARP1 knockdown cells.

To assess directly the flux of RP mRNA translation, we measured the rate of de novo RP protein synthesis. In vivo labeling experiments revealed that knockdown of LARP1 significantly reduced the levels of newly synthesized RP proteins (*Figure 7D*). Furthermore, prolonged LARP1 knockdown decreased RP protein expression (*Figure 7E*) and reduced global protein synthesis (*Figure 7F*) without significantly affecting levels of RP mRNAs. In multiple cancer cell lines (PC3 (prostate), SW40 (colon), HeLa (cervix), and MDA MB231 (breast)), expression levels of LARP1 and RP proteins were well correlated and often enhanced compared to those in non-transformed normal cells (human normal fibroblasts: HNF, and human normal prostate epithelial cells: PrEC) (*Figure 7G*, left upper panels). Furthermore, there was an especially clear trend of higher LARP1 and RP protein expression in multiple prostate cancer cell lines (LnCap, 22RV1, PC3, and Du145) compared with those in non-transformed (PrEC) or transformed (PTN2 and RWPE) normal prostate epithelial cells (left lower panels). Again, there was clear positive correlation between the expression of LARP1 and the RP proteins (e.g. RpS6, right upper panel) but not between LARP1 protein and RpS6 mRNA (right lower panel) in prostate cancer and normal cells. Finally, as previously demonstrated in other cancer cell lines (*Burrows et al., 2010*; *Hopkins et al., 2016*; *Tcherkezian et al., 2014*), the ablation of LARP1 largely blocked cell proliferation in multiple cancer cell lines including prostate cancer cells (*Figure 7—figure supplement 1B*). Together, these observations indicate that LARP1 functions as a phosphorylation-sensitive switch for inhibiting or stimulating the translation of RP mRNAs in response to the mTOR activity (*Figure 7—figure supplement 2*), thereby titrating cellular ribosomes and the capacity of cellular protein synthesis.

## Discussion

In this study, we demonstrate the critical role of LARP1 in the mTORC1-dependent translation of PES-containing mRNAs, especially those that encode ribosome proteins. Key mechanistic insights from this study reveal that (1) LARP1 is a direct substrate of mTOR and S6K1/Akt; (2) the phosphorylation of LARP1 by the mTORC1 pathway triggers the dissociation of LARP1 from the 5′UTR of the RP mRNAs; (3) concomitantly, phosphorylation of LARP1 induces its stable interaction with mTORC1 and recruits mTORC1 to the 3′UTR of RP mRNAs; (4) LARP1 facilitates mTORC1-dependent phosphorylation of 4EBP1 and S6K1, a process essential for inducing translation initiation of RP mRNAs; and (5) while LARP1 is necessary for the blockade of RP mRNA translation in response to starvation, it is required for efficient translation elongation of RP mRNAs. Thus, LARP1 is a unique mTORC1 substrate and regulator, and functions as a molecular switch for turning off or on the translation of RP mRNAs (*Figure 7—figure supplement 2*).

Our data reveal that LARP1 mainly associates with the 3′UTR of mRNAs including RP mRNAs under high mTOR activity conditions, while it also directly binds to PES regions of RP 5′UTRs when mTOR activity is inhibited. Intriguingly, recent reports demonstrate that recombinant LARP1 binds to the TOP sequence of RpL32 and RpS6 in vitro (*Fonseca et al., 2015*; *Lahr et al., 2015*). However, our PAR-CLIP analyses using endogenous LARP1 showed that LARP1 binding within 5′UTRs of

5'TOP mRNAs, such as RpL32 mRNA, occurs predominantly at PES regions in the 3' end of the 5'UTRs and not at the 5'TOP sequences themselves in vivo, when mTORC1 activity is inhibited (*Figure 1C*). We posit that while LARP1 has a strong affinity for pyrimidine cluster sequences, the 5'TOP sequences of RP mRNAs are likely to be occupied by other 5'TOP-binding proteins in vivo. Alternatively, it is possible that mutations or deletions of 5'TOP sequences might structurally affect the function of the PES motifs of RP mRNAs in vitro. Our data demonstrate that the mutation of the PES in the 3' end of the 5'UTR largely reduce LARP1's binding to the 5'UTR of RpL32 mRNA and renders RpL32 mRNA translation resistant to starvation, suggesting that LARP1 binding to the PES regions in the 5'UTRs of RP mRNAs inhibits their translation. Furthermore, the dissociation of LARP1 from the 5'UTR of RpL32 mRNA requires both mTORC1- and S6K1/Akt-dependent LARP1 phosphorylation (*Figure 3D*), indicating that LARP1 phosphorylation by mTORC1 and S6K1/Akt relieves its inhibitory role in translation at the 5'UTRs of RP mRNAs.

Importantly, our data also demonstrate that phosphorylated LARP1 serves as a nucleation site onto which mTORC1 can associate with translationally-competent mRNAs. mTORC1-dependent LARP1 phosphorylation plays a key role in scaffolding mTORC1 on LARP1-interacting RpL32 mRNA (*Figure 5C*). Furthermore, phosphorylated LARP1 and mTORC1 mainly associates with the 3'UTR of RP mRNAs under high mTOR activity conditions. In contrast, phospho-defective mutant LARP1 interacts with the 5'UTR of RP mRNAs regardless of the status of mTOR activity and fails to scaffold mTORC1. Therefore, mTORC1 association with the 3'UTRs of RP mRNAs via LARP1 interaction may allow for a local enhancement of mTORC1 activity to activate the eIF4F complex and thus secure intact translation of LARP1-interacting RP mRNAs. This model provides the mechanistic underpinning for how translation of PES containing mRNAs, such as RP mRNAs, is regulated by LARP1 and may explain why the translation of these mRNAs is sensitive to mTOR activity (*Figure 7—figure supplement 2*).

Given that mTORC1-dependent LARP1 phosphorylation to scaffold mTORC1 is important for mTORC1 phosphorylation of S6K1, mTORC1 may phosphorylate LARP1 prior to S6K1-dependent LARP1 phosphorylation. In support of this model, formation of the LARP1-mTORC1 complex precedes full S6K1 activity/phosphorylation by growth factor/nutrient stimulation (*Figure 4E*). Alternatively, Akt may first phosphorylate LARP1 to prepare mTORC1 phosphorylation of LARP1 for its dissociation from the 5'UTR of RP mRNAs. This functional redundancy of S6K1 and Akt in the phosphorylation of LARP1 may explain previous studies where translation of certain 5'TOP mRNAs was shown to be intact in S6K1/2 null cells but still rapamycin sensitive (*Pende et al., 2004*).

Tcherkezian et al. recently proposed that upon mTOR activation, LARP1 associates with the initiation complex as well as polysomes through PABP (*Tcherkezian et al., 2014*). In addition, they proposed that the DM15 repeats of LARP1 play important roles for its interaction with PABP, eIF4E, and polysomes. Although Tcherkezian et al. and our study both demonstrate important positive roles of LARP1 in the translation of certain 5' TOP mRNAs, our data indicate that the direct binding of LARP1 to a subset of mRNAs does not depend on direct binding to PABP. Instead, we demonstrate that endogenous LARP1 interactions with PABP1 and initiation factors are sensitive to RNase A treatment and thus occur indirectly through binding of common shared mRNA substrates. Intriguingly, our biochemical data suggest that the region containing the DM15 motif is necessary for LARP1 to interact with mTORC1. We speculate that reduced translation in cells expressing mutant LARP1 lacking the complete DM15 motif demonstrated by Tcherkezian et al. may be in part due to a loss of LARP1's scaffolding role for mTORC1, thereby mitigating translation initiation of mRNAs that bind to the mutant LARP1. Moreover, Tcherkezian et al. demonstrated that LARP1 knockdown significantly enhanced the level of 80S monosomes, indicative of a strong inhibition of initiation similar to short-term mTOR inhibitor treatment. They also observed that levels of RP mRNAs associated with polysomes were significantly reduced in LARP1 knockdown cells compared to those in control cells. In contrast, we observed that levels of RP mRNAs in fractions containing polysomes are rather increased in LARP1 knockdown cells, which is consistent with the data more recently published by Fonseca et al., although our data disagreed with the conclusion that LARP1 is a simple suppressor of translation as proposed by *Fonseca et al. (2015)*. While the reasons for these discrepancies remain unclear, our multiple lines of biochemical evidence indicate that loss of LARP1 causes defects in translation at multiple steps, including abnormal translation initiation and inefficient translation elongation likely due to ribosome stalling on LARP1-interacting mRNAs. Therefore, we propose that LARP1 function is context dependent: non-phosphorylated LARP1 acts as a suppressor of translation

initiation of RP mRNAs when mTOR activity is inhibited, whereas upon mTOR activation, phosphorylated LARP1 promotes translation of RP mRNAs at both translation initiation and elongation steps.

Circularization of linear mRNAs through the interaction between the 5'Cap-binding translation initiation factors and PABPs has been recognized as the active conformation for their translation (*Wells et al., 1998*). mTORC1-dependent phosphorylation of 4EBPs plays a critical role in the induction of this topological change. Our characterization of LARP1 interaction with RpL32 mRNA indicates that LARP1 interacts with both the 5' and 3'UTR of RpL32 mRNA under low mTOR activity conditions. This suggests that certain RP mRNAs that interact with LARP1 under low mTOR activity conditions may form an inactive circular conformation (*Figure 7—figure supplement 2*). Although visualization or/and stoichiometric analysis of the LARP1-RP mRNA complex will be necessary to elucidate this proposed inactive circular conformation, it is reasonable to speculate that retaining such a circular conformation with LARP1 may provide an efficient way to rapidly toggle between translationally on and off states.

## Matherials and methods

### Antibodies

Antibodies were purchased from the following sources: Antibodies to mTOR (cat. # 2983, RRID:AB_2105622 for western blotting and immunostaining), Raptor (cat. # 2280, RRID:AB_561245), pT389 S6K1 (cat. # 9206, RRID:AB_2285392), S6K (cat. # 9202, RRID:AB_823592), pS473 Akt (cat. # 9270, RRID:AB_329824), Akt (cat # 9272, RRID:AB_329827), phospho-Akt Substrate (RXXS/T) (cat. # 9614, RRID:AB_2225188), pT37/46 4EBP1 (cat. # 2855, RRID:AB_560835), 4EBP1 (cat. # 9644, RRID:AB_10691384), S6 (cat. # 2217, RRID:AB_331355), pS235/236 S6 (cat. # 4856, RRID:AB_2181037 for western blotting), pS240/244 S6 (cat. # 5364, RRID:AB_10694233 for co-IP), RpL13a (cat. # 2765, RRID:AB_916223), PABP1 (cat # 4992, RRID:AB_2156887), mLST8 (cat. # 3274, RRID:AB_823685), PRAS40 (cat. # 2610, RRID:AB_916206), pULK1 (cat. # 6888, RRID:AB_10829226), and ULK1 (cat. # 6439, RRID:AB_11178933) from Cell Signaling Technology; antibodies to LARP1 (cat. # A302-087A, RRID:AB_1604274 for co-IP and western blotting, cat. # A310-088A, RRID:AB_2632225 for IP and western blotting and cat. # IHC-00559, RRID:AB_10631280 for immunostaining), Rictor (cat. # A300-458A, RRID:AB_420924), eIF4G (cat. # A301-776A, RRID:AB_1211011), eIF3D (cat. # A301-758A, RRID:AB_1210970), and eIF3B (cat. # A301-761A, RRID:AB_1210995) from Bethyl Laboratories; antibodies to eIF4E (cat. # sc-9976, RRID:AB_627502) and mTOR/FRAP (cat. # sc-1549, RRID:AB_631981 for co-IP) from Santa Cruz Biotechnology; antibodies to β-Actin (cat. # A1978, RRID:AB_476692), and Flag M2 antibodies (cat. # F1804, RRID:AB_262044) from Sigma-Aldrich; antibodies to HA tag (cat. # MMS-101P, RRID:AB_2314672), and Myc tag (cat. # MMS-150P, RRID:AB_291322) from Covance; antibody to LAMP2 (cat. # H4B4, RRID:AB_528129) from the Developmental Studies Hybridoma Bank at the University of Iowa; HRP-conjugated mouse secondary antibody (cat. # NA931-1ML, RRID:AB_772210) and HRP-conjugate rabbit secondary antibody (cat. # NA934-1ML, RRID:AB_772206) from GE Healthcare; Alexa fluor 594 goat anti-rabbit IgG (cat. # A11012, RRID: AB_141359) and Alexa fluor 488 goat anti-mouse IgG (cat. # A11001, RRID:AB_2534069) from Invitrogen.

### Cell culture and treatment

HEK293T and HEK293E (a generous gift from Dr. Diane Fingar, University of Michigan) cells (*Tee et al., 2002*) were cultured in DMEM with 10% fetal bovine serum (FBS) and 100 units/ml penicillin/streptomycin (Invitrogen). Human breast cancer cell line: MDA-MB-231 cells (kindly provided by Dr. Shaomeng Wang, University of Michigan) (*Lu et al., 2008*), human colon cancer cell line: SW480 cells (kindly provided by Dr. Eric Fearon, University of Michigan) (*Mazzoni et al., 2015*), and human prostate cancer cell lines: LNCaP, 22RV1 and PC3 and benign prostate cell lines: PNT2 and RWPE (generous gifts from Dr. Arul Chinnaiyan, University of Michigan) (*Ateeq et al., 2011*; *Helgeson et al., 2008*; *Tomlins et al., 2007*) were cultured in RPMI 1640 with 10% FBS and 100 units/ml penicillin/streptomycin and RWPE cells were cultured in keratinocyte serum-free medium (Invitrogen) with Supplements for Keratinocyte-SFM (Invitrogen). Primary prostate cell line: PrEC was purchased from LONZA and cultured in the PrEGM media (LONZA). Each cell line was tested for mycoplasma contamination and confirmed the absence of mycoplasma using fluorescence- and

PCR-based methods (Invitrogen) before subjecting to the experiments. In order to inhibit kinases involved in the PI3K pathway we treated cells with 250 nM Torin-1, 20 µM PF 470861 (Tocris bioscience), 2.5 µM PP242 (Sigma-Aldrich), 100 µM rapamycin (LC laboratories), 1 µM Akt1/2 inhibitor (Sigma-Aldrich), or 2 µM MK-2206 (Active BioChem) for indicated times. For growth factors/nutrients starvation, cells were incubated with either HBSS (Hank's Balanced Salt Solution, Invitrogen) or DPBS (Dulbecco's Phosphate-Buffered Saline, Invitrogen) for the indicated times. For stimulation of these, growth media were added for the indicated times.

## Cell lysis, immunoprecipitation and immunostaining

Cells were harvested with CHAPS lysis buffer (40 mM HEPES [pH 7.5], 120 mM NaCl, 1 mM EDTA, 0.3% CHAPS, and 20 mM glycerophosphate, 10 mM NaF, 10 mM sodium pyrophosphate [Sigma-Aldrich] and EDTA-free protease inhibitor [Roche]) by incubating on ice for 15 min. The soluble fractions were isolated by centrifugation at 14,000 rpm for 15 min at 4°C. For immunoprecipitation, 1 µg of antibodies was added and incubated with gentle rocking for 3–6 hr at 4°C. 20 µl of 50% slurry of protein G sepharose (GE Healthcare) was added and incubated for additional 1 hr. Immunoprecipitates were washed with CHAPS lysis buffer for five times and then denatured at 100°C for 5 min in 1X SDS sample buffer. For RNase A treatment, lysates were incubated with 20 µg of RNase A (Affymetrix) for 1 hr at room temperature with gentle rocking. After spinning at 14,000 rpm at 4°C for 15 min to remove protein aggregates, remaining supernatant was applied for further analysis. For immunostaining, cells on round cover slips (Fisher Scientific) were fixed with 4% paraformaldehyde for 20 min and washed with PBS three times. Fixed cells were permeabilized with 0.2% TX-100 for 10 min at room temperature and blocked in 2% BSA in PBS for 1 hr. After washing with PBS three times, cells were incubated with indicated primary antibodies for three hours at room temperature and washed with PBS three times. Cells were incubated with Alexa fluor 488/594 goat anti-mouse/rabbit IgG for 1 hr at room temperature. Cells were washed with PBS three times and once with water. The cover slips were mounted on the glass slides using ProLong Gold antifade reagent with DAPI (Life technologies) and imaged with 63X oil-immersion objective by using a Leica TCS SP5 confocal microscope.

## Capture and sequencing of LARP1-bound RNA fragments using PAR-CLIP-seq

To label cells with 4-thiouridine (4-SU), cells were seeded in two 15 cm plates to grow overnight to reach 70% confluency. On the next day, cells were incubated with 100 µM 4-SU for 14 hr. After washing with cold PBS, cells were irradiated in the CL-1000 ultraviolet Crosslinker (UVP) on ice with 150 mJ/cm$^2$. Cross-linked cell pellets were collected by scraping and lysed with 600 µl PAR-CLIP lysis buffer (0.1% SDS, 0.5% deoxycholate, 0.5% NP-40 in PBS without Mg$^{2+}$, Ca$^{2+}$) on ice for 10 min. To remove DNA, 10 µl of RQ1 DNAse was added into each tube and tubes were incubated at 37°C for 10 min with gentle rocking. Afterward, 1 U/µl of RNase T1 (Fermentas) was added and the lysates were incubated at room temperature for 15 min with gentle rocking. Lysates were spun at 4°C for 10 min at 14,000 rpm. The soluble fractions were incubated with LARP1 antibody-Dynabeads Protein A for 1 hr at 4°C. Immunoprecipitates were collected on a magnetic stand and were then washed three times with IP-wash buffer (50 mM HEPES-KOH [pH 7.5], 300 mM KCl, 0.05% NP-40, 0.5 mM DTT, and EDTA-free protease inhibitor). Immunoprecipitates were resuspended in 40 µl of IP wash buffer containing 50 U/µl RNase T1 (Fermentas) and incubated at room temperature with gentle rocking for 15 min followed by incubation on ice for 5 min. Immunoprecipitates were washed with high-salt wash buffer (50 mM HEPES-KOH [pH 7.5], 500 mM KCl, 0.05% NP-40, 0.5 mM DTT, and EDTA-free protease inhibitor), three times with the PAR-CLIP lysis buffer, and twice with high-salt wash buffer (0.1% SDS, 0.5% deoxycholate, 0.5% NP-40 in 5X PBS without Mg$^{2+}$, Ca$^{2+}$) followed by washing twice with PNK (polynucleotide kinase) buffer (50 mM Tris-Cl pH 7.4, 10 mM MgCl$_2$, 0.5% NP-40). For visualization of crosslinked RNAs, immunoprecipitates were incubated in 40 µl of the PNK mixture (1 µl of P$^{32}$γATP, 4 µl of 10X PNK buffer [NEB], 2 µl of T4 PNK enzyme [NEB], 33 µl of water) for 30 min at 37°C. Labeled immunoprecipitates were washed three times with PNK buffer and resuspended with 30 µl of 2X NuPAGE LDS sample buffer (15 µl of 1 X PNK with 15 µl of Novex LDS sample buffer). Denatured samples were resolved in 4–12% NuPAGE Bis-Tris gel and transferred to nitrocellulose membrane at 30 V for 1 hr using NuPAGE transfer buffer. The membrane was exposed

to X-ray film at −80°C for 4 hr to visualize crosslinked RNAs. Molecular biology procedures for cloning LARP1-bound RNA fragments was described previously (*Freeberg et al., 2013*).

## Sequence read processing

PAR-CLIP-seq and mRNA-seq reads were processed to remove linkers. All reads were mapped to the human transcriptome version GRCh37 using Bowtie (*Langmead et al., 2009*) allowing for up to three mismatches with the following parameters: -v 3 k 100 –best –strata –phred33-quals. mRNA-seq reads mapping perfectly to the transcriptome were kept; reads mapping perfectly to multiple loci were distributed evenly among the mapped positions. Transcript RPKM values were calculated as the number of reads per million mapped reads aligning to a transcript normalized to transcript length in kilobases. Replicate mRNA-seq libraries had a high Pearson correlation coefficient ($R^2$ = 0.99975), so transcript RPKM values were averaged from the two libraries. PAR-CLIP-seq reads with 0–2 T-to-C mismatches were clustered into peaks with at least one overlapping nucleotide. Clusters were smoothed with a Gaussian smoothing technique as described in (*Freeberg et al., 2013*) Clusters with at least one read containing a T-to-C conversion event were kept as LARP1 binding sites, and all reads containing 0–2 T-to-C conversion events were summed per binding site and normalized to the number of million mapped reads per library (RPM). The RPM of binding sites located in genic regions were additionally normalized to the gene RPKM and multiplied by 1000 to account for the kilobase normalization of gene RPKM values.

## Gene ontology term enrichment analysis

GO term enrichment analysis was performed using the topGO Bioconductor package for R. The background gene list was restricted to genes with reads in at least one of our replicate mRNA-seq libraries. The Fisher's exact test was used to measure the significance of enriched GO terms, and p-values were corrected for multiple testing using the Bonferroni correction method. GO terms with adjusted p-values<0.001 were submitted to REVIGO (*Supek et al., 2011*) which summarizes long lists of GO terms (accompanied by enrichment p-values) by removing redundant terms and grouping terms into larger categories. REVIGO parameters were: allowed similarity = 0.9, database with GO term sizes: Homo sapiens, semantic similarity: SimRel. From these REVIGO-derived categories, GO terms were manually grouped into the following categories: translation, cell differentiation and development, protein localization to the ER, regulation of signaling response, response to stimulus, antibody production, metabolism, and other (*Supplementary file 2*). For example, the categories 'translational elongation', 'translational initiation', and 'translation' were all grouped into the 'translation' super-category. Genes are considered 'translation-related' if they are annotated to at least one of the GO terms in our translation super-categry.

## Identification of sequence motifs

Sequence motifs were searched for within the 198 and 186 5'UTR LARP1 binding sites found under growing and mTOR inactive conditions, respectively, using MEME with default parameters (*Bailey et al., 2009*) Binding site sequences were extended by 15nt up- and down-stream for this search. Additionally, sequence motifs were searched for within CDS and 3'UTR LARP1 binding sites under growing and mTOR inactive conditions using the same parameters.

## Measurement of newly synthesized protein (Click-IT labeling)

Cells were seeded in 6-well plates a day before labeling to reach 70% confluency on the day of labeling. Cells were washed twice with warm methionine/cysteine free DMEM with 1 x L-glutamine (Invitrogen) and incubated with methionine/cysteine free DMEM with 1 X L-glutamine and 10% dialyzed FBS (Invitrogen) for 1 hr. Cells were labeled with 25 μM Click-IT AHA (L-Azidohomoalanine) for 4 hr in a $CO_2$ incubator. Labeled cells were washed once with PBS and lysed with NP-40 lysis buffer (10 mM Tris-Cl [pH 7.5], 2 mM EDTA, 100 mM NaCl, 1% NP-40, 50 mM NaF, 10 mM sodium pyrophosphate and EDTA-free protease inhibitor) for 10 min on ice. The soluble fractions isolated by centrifugation were transferred into new tubes and the Click-IT reaction was performed in 1 X Click-IT reaction buffer according to the manufacturer's guide. The Click-IT reaction was stopped by addition of SDS sample buffer and denatured by boiling. Samples were resolved by SDS-PAGE and transferred to PVDF membrane for immunoblot analysis. Labeled proteins were probed with Avidin-HRP

(BioRad) and membranes were visualized by ECL on x-ray film. Intensity of signal was quantified by Image J.

## Polysome analysis

Prior to lysis, cells were treated with 100 µg/mL cycloheximide (CHX) for 5 min and washed with cold PBS containing 100 µg/mL CHX. Cells were harvested in 1 ml of cold PBS containing 100 µg/mL CHX. Cell pellets were resuspended in hypotonic buffer (5 mM Tris [pH 7.5], 2.5 mM $MgCl_2$, 1.5 mM KCl, 0.3% CHAPS, 20 mM glycerophosphate, 10 mM NaF, 10 mM sodium pyrophosphate and EDTA-free protease inhibitor) containing 100 µg/mL CHX, 2 mM DTT, and 200 U/ml RNasin (Promega) by vortexing for 5 s. Supernatants were collected by centrifugation for 10 min at 14,000 rpm, 4°C, and the $A_{260}$ of lysates was measured to normalize RNA levels. Lysates of 30 O.D.$_{260}$ were loaded on 10–50% sucrose density gradients (100 mM HEPES [pH 7.6], 1M KCl, 50 mM $MgCl_2$, 100 µg/mL CHX, 200 U/ml RNasin, and EDTA-free protease inhibitor) and centrifuged at 35,000 rpm for 3 hr at 4°C. Gradients were fractionated and the optical density at 254 nm was recorded using an ISCO fractionator (Teledyne ISCO). For immunoblotting, the fractions were concentrated by Vivaspin concentrator (Sartorius). For quantitative PCR normalization, 5 ng of luciferase control mRNA (Promega) was added to each fraction. RNA was extracted using TRIzol and cDNA was synthesized by SuperScript III First-Strand Synthesis System (Invitrogen) following the manufacturer's instructions.

## In vitro kinase assay

For mTOR in vitro kinase assays, HEK293T cells were lysed with CHAPS lysis buffer and endogenous mTOR, Flag-mTOR Wt (wild type), or Flag-mTOR KD (kinase defective) was immunoprecipitated and subjected to in vitro kinase assays. For LARP1 phosphorylation, the kinase reaction was performed in kinase reaction buffer (25 mM HEPES [pH 7.4], 50 mM KCl, 10 mM MgCl2, 200 mM cold ATP and 5 µCi of $^{32}$PγATP) using 2 µg of GST-LARP1 fragment (654–731 aa) purified from bacteria. For 4EBP1 phosphorylation, 200 ng of GST-4EBP1 (full length) was used as a substrate for the kinase reaction without using $^{32}$PγATP, and the phosphorylation of 4EBP1 was detected by the indicated phospho-specific 4EBP1 antibody. For AKT and S6K1 kinase assays, AKT-HA, AKT-HA KD, HA-S6K1 F5A 3DE (constitutive active), or HA-S6K1 KD was immunoprecipitated from HEK293T cells and kinase reactions were performed in the AKT/S6K1 kinase reaction buffer (75 mM Tris-Cl [pH 7.5], 15 mM $MgCl_2$, 1.5 mM DTT, 1.5 µg/ml BSA, 200 mM cold ATP and 5 µCi of $P^{32}$γATP) using 2 µg of GST-LARP1 fragments (722–822 aa or 929–1019 aa).

## shRNAs, lentivirus production, and stable knock-down

Short-hairpin RNAs (shRNAs) against LARP1 or eIF4G1 were cloned into the lentivirus plasmid pLKO.1-puro. HEK293T cells were transfected with pLKO.1-puro cloned with shRNA, psPAX2 (packaging plasmid) and pMD2 (envelope plasmid) using the calcium phosphate method. Lentivirus-containing supernatants were collected and spun at 23,000 rpm for 90 min. Virus pellets were resuspended with Opti-MEM (Invitrogen). Cells were infected for 24 hr and selected with 2 µg/ml puromycin for additional 24 hr. 72 hr post-infection, stably knocked-down cells were harvested and processed for further analysis.

shRNA scramble (control):
Sense:
5'CCGGCCTAAGGTTAAGTCGCCCTCGCTCGAGCGAGGGCGACTTAACCTTAGGTTTTTC3'
Antisense:
5'AATTGAAAAACCTAAGGTTAAGTCGCCCTCGCTCGAGCGAGGGCGACTTA
ACCTTAGG3' shRNA LARP1 UTR-2:
Sense:
5'CCGGGGTGAGGACTTCATCTCAACACTCGAGTGTTGAGATGAAGTCCTCACCTTTTTC3'
Antisense:
5'AATTGAAAAAGGTGAGGACTTCATCTCAACACTCGAGTGTTGAGATGAAGTCCTCACC3'
shRNA LARP1-3
Sense:
5'CCGGGCCAGTCTCAGGAGATGAACACTCGAGTGTTCATCTCCTGAGACTGGCTTTTTC3'
Antisense:

5'AATGAAAAAGCCAGTCTCAGGAGATGAACACTCGAGTGTTCATCTCCTGAGACTGGC3'
shRNA eIF4G1-1
Sense:
5'CCGGGGATCCCACTAGACTACAAGGCTCGAGCCTTGTAGTCTAGTGGGATCCTTTTTC3'
Antisense:
5'AATTGAAAAAGGATCCCACTAGACTACAAGGCTCGAGCCTTGTAGTCTAGTGGGATCC3'

## Oligonucleotides used for qPCR

RpS6:
Forward: 5'TGTCCGCCTGCTACTGAGTAA3'
Reverse: 5'GCAACCACGAACTGATTTTCTC3'
RpS3A:
Forward: 5'AGGGTCGTGTGTTTGAAGTGA3'
Reverse: 5'CATGGAAGTTAGTCAGGCAGTTT3'
RpS18:
Forward: 5'GCGGGAGAACTCACTGAGG3'
Reverse: 5'CGTGGATTCTGCATAATGGTGAT3'
RpL26:
Forward: 5'GACTTCCGACCGAAGCAAGAA3'
Reverse: 5'TGCACCCGTTCAATGTAGATAAC3'
RpL32:
Forward: 5'GCCCAAGATCGTCAAAAAGAGA3'
Reverse: 5'TCCGCCAGTTACGCTTAATTT3'
Actin:
Forward: 5'CATGTACGTTGCTATCCAGGC3'
Reverse: 5'CTCCTTAATGTCACGCACGAT3'
GAPDH:
Forward: 5'TTGCCATCAACGACCCCTTC3'
Reverse: 5'TTGTCATGGATGACCTTGGC3'
Firefly luciferase:
Forward: 5'CTCACTGAGACTACATCAGC3'
Reverse: 5'TCCAGATCCACAACCTTCGC3'

## Oligonucleotides used for generating pGL3-RpL32 5'conv

Forward: 5'gcctacggaggtggcag**GGTAG**tccttctcggcatc3'
Reverse: 5'gatgccgagaagga**CTACC**ctgccacctccgtaggc3'

## Statistical analysis

Data are representative of at least two independent experiments. All values were given as means±SEM from three independent biological replicates. Comparisons were performed using Student's t-test or one-way factorial ANOVA followed by Bonferroni's post-hoc analysis.

## Acknowledgement

We thank Arul Chinnaiyan and Shaomeng Wang for providing breast and prostate cancer cell lines, respectively, Eric Fearon for helpful advice, Ivan Topisirovic for instructions on polysome fractionation, Aristotle Mannan for technical assistance, and Philip Gafken (Fred Hutchinson Cancer Research Center) for proteomic analyses. This work was supported by grants from the NIH (DK083491/ GM110019 to KI, GM088565 to JKK), DOD (TS140055 to KI), and ACS Research Scholar Grant (to JKK) and the Ruddon Research Fund in Cancer Biology (to KI).

## Additional information

### Funding

| Funder | Grant reference number | Author |
| --- | --- | --- |
| National Institute of General Medical Sciences | GM088565 | John K Kim |
| National Institute of General Medical Sciences | GM110019 | Ken Inoki |
| U.S. Department of Defense | TS140055 | Ken Inoki |
| National Institute of Diabetes and Digestive and Kidney Diseases | DK083491 | Ken Inoki |

The funders had no role in study design, data collection and interpretation, or the decision to submit the work for publication.

### Author contributions

SH, Resources, Formal analysis, Validation, Investigation, Writing—original draft; MAF, Software, Formal analysis, Methodology, Writing—original draft, Writing—review and editing; TH, Formal analysis, Investigation, Methodology; AK, Investigation, This author contributed to clone all the cDNAs of LARP family proteins and LARP1 mutants, which was essential to generate specific figures (e.g. Fig. 5B) and other figures using LARP1 mutants; YY, Formal analysis, Investigation, Visualization; TF, Investigation, This author also contributed to clone cDNAs for LARP1 mutants, which were essential to generate specific figures (e.g. Fig. 5A, 5G, and 5H); TS, Investigation, This author originally identified LARP1 as a mTORC1 interacting protein using proteomics approaches (data not shown), which was essential part of work for initiating this project; JKK, Supervision, Funding acquisition, Validation, Writing—review and editing; KI, Supervision, Funding acquisition, Validation, Writing—original draft, Writing—review and editing

### Author ORCIDs

Ken Inoki, http://orcid.org/0000-0001-8882-444X

## Additional files

### Supplementary files

• Supplementary file 1. LARP1 binding sites.

• Supplementary file 2. Gene ontology terms enriched in LARP1-bound genes under growing and mTOR inactive conditions.

• Supplementary file 3. LARP1-bound genes classified as non-TR, TR, and RP-encoding genes.

• Supplementary file 4. Location of 5'TOP/5'TOP-like and LARP1 binding site in select 5'UTRs.

### Major datasets

The following dataset was generated:

| Author(s) | Year | Dataset title | Dataset URL | Database, license, and accessibility information |
| --- | --- | --- | --- | --- |
| Hong S, Freeberg MA, Avani K, Ting H, Yao Y, Tomoko F, Andy K, Tsukasa S, Kim JK, Inoki K | 2014 | LARP1 mRNP scaffolds mTORC1 to stimulate translation initiation of an essential class of mRNAs | https://www.ncbi.nlm.nih.gov/geo/query/acc.cgi?acc=GSE59599 | Publicly available at the NCBI Gene Expression Omnibus (accession no: GSE59599) |

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
