## [Decision Letter]

Thank you for submitting your article "LARP1 functions as a molecular switch for mTORC1-mediated translation of an essential class of mRNAs" for consideration by *eLife*. Your article has been favorably evaluated by Tony Hunter (Senior Editor) and three reviewers, one of whom is a member of our Board of Reviewing Editors. The reviewers have opted to remain anonymous.

The reviewers have discussed the reviews with one another and the Reviewing Editor has drafted this decision to help you prepare a revised submission.

Summary:

In this elegant study, the authors characterize the role of LARP1 in the regulation of mRNA translation, specifically assessing TOP mRNAs. The role of LARP1 in the translation of TOP mRNAs has been a contentious issue, and the authors bring novel data suggesting a dual function for LARP1, which they describe as a phosphorylation-sensitive molecular switch for mRNA translation and ribosome biogenesis. Previous reports have shown both positive and negative functions for LARP1 in TOP mRNA translation. Hong et al. suggest that LARP1 binds the 5'UTR of translation-related mRNAs to repress their translation during nutrient starvation. However, LARP1 phosphorylation by mTOR/S6K decreases this binding to promote 3'UTR binding and enhance translation. They show that LARP1 acts as an mTORC1 scaffold to facilitate mTORC1-dependent phosphorylation of S6K and 4E-BP1. Together, these data suggest that LARP1 functions as a molecular switch that senses mTORC1 activity and act as an effector.

This manuscript provides very insightful results that increase our understanding of the function of LARP1.

Some modifications and/or clarifications would be required prior to publication:

Essential revisions:

1) Although the manuscript is well done the main conclusion that LARP1 inhibits or facilitates translation depending on where it binds on the mRNA has not been fully demonstrated. The experiments shown in Figure 3 are not compelling because different mRNAs containing different 5' or 3' sequences are being compared. Ideally the authors should design a luciferase reporter containing RpL32 5' and 3' UTRs and introduce mutations in the LARP1 binding sites rather than deleting the entire UTRs, so that the mRNAs have all the same length. Alternatively, the authors could test luciferase activity with the same reporter in cells depleted of LARP1 and then perform complementation assays with LARP1 mutants that can or cannot be phosphorylated. Related to this point: In Figure 3, have the authors verified mRNA levels to ensure that what is being measured is not simply changes in mRNA stability?

2) The authors repeatedly state that LARP1 is a substrate for both S6K1 and Akt. However, Akt as a kinase for LARP1 is only shown by in vitro kinase assays and overexpression of a constitutively active Akt in cells – both are artifact-prone. In fact, Rap or PP242 abolishes pS979-LARP1 in cells (Figure 2), which suggests that at least for S979 Akt is not involved in vivo. While Akt may be considered by the authors to potentially explain the behavior of the S6K1/2 double KO cells (Pende et al., 2004), it is not justified to include Akt in the conclusions of the current manuscript unless more direct evidence is provided, such as treatment of cells with an Akt inhibitor or RNAi.

3) It is not clear why the authors used Torin instead of rapamycin in the reporter assays (Figure 3), and total mTOR instead of mTORC1 or mTORC2 for the kinase assays (Figure 2). It seems obvious from early on that mTORC1, and not mTORC2, is responsible based on the sensitivity to rapamycin (Figure 2) and amino acids (Figure 2). Also, the S6K1 inhibitor PF could have been included in the reporter assays (Figure 3) to distinguish the role of mTORC1 and S6K1. There shouldn't be any technical hurdle. Perhaps the authors did perform those experiments and had a reason not to include the data?

4) What is the proposed mechanism by which LARP1 phosphorylation decreases 5'UTR binding and increases mTORC1 and 3'UTR binding? Have the authors tested phospho-mimetic mutants? Related to this point: The model in Figure 7—figure supplement 2 showing LARP bridging the 5' and 3' UTRs of an mRNA is not supported by any experimental evidence. Do the authors have any evidence that LARP1 can bind simultaneously two mRNAs or that LARP1 dimerizes?

---

## [Author Response]

*Essential revisions:*

*1) Although the manuscript is well done the main conclusion that LARP1 inhibits or facilitates translation depending on where it binds on the mRNA has not been fully demonstrated. The experiments shown in Figure 3 are not compelling because different mRNAs containing different 5' or 3' sequences are being compared. Ideally the authors should design a luciferase reporter containing RpL32 5' and 3' UTRs and introduce mutations in the LARP1 binding sites rather than deleting the entire UTRs, so that the mRNAs have all the same length. Alternatively, the authors could test luciferase activity with the same reporter in cells depleted of LARP1 and then perform complementation assays with LARP1 mutants that can or cannot be phosphorylated. Related to this point: In Figure 3, have the authors verified mRNA levels to ensure that what is being measured is not simply changes in mRNA stability?*

In response to the reviewer’s suggestions, we examined the roles of LARP1 phosphorylation on its mRNA binding and in the regulation of mRNA translation using the wild type or mutant RpL32 mRNA reporters containing both 5’ and 3’ UTRs. Although our RIP assays using either the 5’ or 3’ UTR reporter of RpL32 mRNA showed dynamic changes in the affinity for LARP1 binding (Figure 3, and E), under in vivo crosslinking conditions (PAR-CLIP), LARP1 constitutively and equally interacts with both the wild type 5’/3’ and the mutant 5’conv/3’ reporters regardless of cellular mTOR activity (new Figure 3 in the revised manuscript). These observations are consistent with the data in our transcriptome-wide PAR-CLIP analyses: LARP1 interacts with the 3’UTRs of RP mRNAs under both mTOR active and inactive conditions, while LARP1 also interacts with the 5’ UTRs, in addition to the 3’UTRs of these mRNAs, under mTOR inactive conditions. Furthermore, LARP1 constitutively associates with PABP1 in an RNAse A-sensitive manner regardless of cellular mTOR activity (Figure 4). Together, these observations suggest that LARP1 phosphorylation does not induce its complete dissociation from RpL32 mRNA. Rather, LARP1 phosphorylation (i.e., mTOR active condition) induces the dissociation of the 3’UTR-bound LARP1 from the 5’UTR of RpL32 mRNA.

To further address the critical point raised by the reviewer, we performed luciferase assays using the RpL32 reporter, which contains full length 5’/3’ or 5’conv/3’ UTRs in cells exclusively expressing the wild type or 4A LARP1 mutant. For this experiment, endogenous LARP1 was first ablated by shRNAs, and the shRNA-resistant Myc-LARP1_resist_ or Myc-LARP1_resist_ 4A mutant was co-transfected with the indicated reporters in HEK293T cells. Levels of luciferase activity were monitored under growth cell culture conditions. As performed in the original Figure 3 (now Figure 3 in the revised manuscript), the luciferase activity was also normalized by luciferase mRNA levels determined by qPCR.

The data demonstrated that phosphorylatable wild type Myc-LARP1_resist_ has some advantages in stimulating translation of the 5’conv/3’ reporter compared to the wild type 5’/3’ reporter. In contrast, the phopho-defective Myc- LARP1_resist_ 4A mutant significantly and equally inhibits the translation of both reporters, regardless of the existence (5’/3’) or absence (5’conv/3’) of the LARP1 binding site in the 5’UTR. These observations indicate that the dissociation of LARP1 from the PES motif in the 5’UTR and the phosphorylation of LARP1, which enhances both LARP1 binding to the 3’UTR and to mTORC1, are important processes in stimulating RpL32 mRNA translation. Note that the translation of the 5’conv/3’ mutant reporter was slightly, but significantly, enhanced compared to the 5’/3’ wild type reporter in cells expressing ectopic Myc-LARP1_resist_, which was not seen in the cells expressing just endogenous LARP1 (Figure 3). We speculate that overexpressed Myc-LARP1 may not be fully phosphorylated in our experimental conditions. As a consequence, non-phosphorylated wild type LARP1 may still bind to the PES in the 5’UTR and reduce the translation of the wild type 5’/3’ reporter. These new data have been included as new Figure 3 in the revised manuscript.

*2) The authors repeatedly state that LARP1 is a substrate for both S6K1 and Akt. However, Akt as a kinase for LARP1 is only shown by in vitro kinase assays and overexpression of a constitutively active Akt in cells – both are artifact-prone. In fact, Rap or PP242 abolishes pS979-LARP1 in cells (Figure 2), which suggests that at least for S979 Akt is not involved in vivo. While Akt may be considered by the authors to potentially explain the behavior of the S6K1/2 double KO cells (Pende et al., 2004), it is not justified to include Akt in the conclusions of the current manuscript unless more direct evidence is provided, such as treatment of cells with an Akt inhibitor or RNAi.*

The reviewer raises an important question. Our original data suggested that Akt and S6K1 may function redundantly to phosphorylate the S770/S979 sites of LARP1 that are required for its dissociation from the 5’UTR and subsequent tight association with the 3’UTR of RpL32 mRNA. We showed that mTORC1 associates with the 3’UTR of RpL32 mRNA in a manner dependent on phosphorylated LARP1 (Figure 5) and proposed that this configuration may play a key role in initiating mTORC1-dpendent 4EBP1 and S6K1 phosphorylation. Therefore, it is plausible that Akt is the responsible kinase initially phosphorylates S770/S979 of LARP1 and then S6K1 takes over that role in phosphorylating these residues to sustain the configuration that is important RpL32 mRNA translation.

To demonstrate that Akt is responsible for phosphorylating S770/S979 of LARP1 in a physiological in vivo context, we examined the phosphorylation status of LARP1 under acutely stimulated (growth factor/amino acid starvation for overnight/60 min, respectively, then stimulation with growth factor/amino acid for 10 min) or steady state growth conditions with or without the addition of a specific Akt inhibitor, MK-2206.

As shown in the left panel of the new Figure 2 in the revised manuscript, upon growth factor/amino acid stimulation, the phosphorylation of both Akt and S6K1 (lane 3) was enhanced compared to those under starvation conditions (lane 2). Simultaneously, the phosphorylation of LARP1, as detected by both the Akt substrate antibody and the pS979 LARP1 antibody, was enhanced. Importantly, under this acutely stimulated condition, rapamycin, which completely inhibits S6K1, but not Akt, had little effect on S770/S979 phosphorylation of LARP1. In contrast, the Akt inhibitor MK-2206, as well as Torin1, which inhibit both Akt and S6K1, largely inhibited S770/S979 phosphorylation of LARP1.

Under the steady state growth condition (new Figure 2 in the revised manuscript, right panel), inhibition of S6K1 with rapamycin or PF 4708671, a S6K inhibitor, equally and significantly decreased LARP1 phosphorylation (lane 3 and 6), and Torin1 and MK-2206 further blocked these phosphorylations (lane 4 and 5). Taken together, these observations indicate that Akt is indeed a physiologically relevant primary kinase for S770/S979 phosphorylation of LARP1 especially under acute stimulatory conditions. Under steady state conditions, S6K plays a major role while Akt also partially contributes to the phosphorylation of LARP1. These data are now included as new Figure 2 in the revised manuscript.

*3) It is not clear why the authors used Torin instead of rapamycin in the reporter assays (Figure 3), and total mTOR instead of mTORC1 or mTORC2 for the kinase assays (Figure 2). It seems obvious from early on that mTORC1, and not mTORC2, is responsible based on the sensitivity to rapamycin (Figure 2) and amino acids (Figure 2). Also, the S6K1 inhibitor PF could have been included in the reporter assays (Figure 3) to distinguish the role of mTORC1 and S6K1. There shouldn't be any technical hurdle. Perhaps the authors did perform those experiments and had a reason not to include the data?*

In response to the reviewer’s question, we examined the effect of rapamycin or PF4708671 on the interaction between endogenous LARP1 and the 5’ or 3’ UTR of RpL32 mRNA reporter. As expected, rapamycin or Torin1 treatment produced the same effect on the interaction of LARP1 with the 5’ and 3’UTRs of RpL32 mRNA: rapamycin enhanced the affinity of LARP1 for the 5’UTR but suppressed LARP1’s binding to the 3’UTR under steady state growth conditions.

However, PF4708671, a specific S6K inhibitor, had little effect on LARP1’s binding to the 3’UTR, although PF4708671 did enhance LARP1’s binding to the 5’UTR, as does rapamycin, Torin1, and amino starvation. These data are consistent with the previous observations in our original Figure 3, which showed that dephosphorylation of either S6K1 or mTOR sites of LARP1 is sufficient for LARP1 to interact with the 5’UTR of RpL32 mRNA. However, dephosphorylation of both S6K- and mTOR-dependent phosphorylation sites is required for LARP1 dissociation from the 3’UTR. These new data are now added in the revised manuscript as new Figure 3

*4) What is the proposed mechanism by which LARP1 phosphorylation decreases 5'UTR binding and increases mTORC1 and 3'UTR binding? Have the authors tested phospho-mimetic mutants? Related to this point: The model in Figure 7—figure supplement 2 showing LARP bridging the 5' and 3' UTRs of an mRNA is not supported by any experimental evidence. Do the authors have any evidence that LARP1 can bind simultaneously two mRNAs or that LARP1 dimerizes?*

How LARP1 phosphorylation by Akt/S6K1 and mTOR triggers its dissociation from the 5’UTR but keeps the LARP1-mTORC1 complex on the 3’UTR of mRNA remains unclear. LARP1 possesses two RNA binding motifs (La and RRM-like). La protein (LARP3), which also contains La and two RRMs, has been shown to interact with 5’TOP mRNAs (Intine et al. Mol Cell 2003), suggesting that one or both of these RNA binding motifs may play a role in LARP1 association with the 5’TOP structure. However, recent structural studies demonstrated that the LARP1 DM15 motif, which is specific to LARP1/2 (Figure 5—figure supplement 1) and is positioned adjacent to the Raptor binding and mTOR-dependent phosphorylation sites on LARP1, also interacts with the 5’ cap structure as well as the TOP sequence of RpS6 mRNA in vitro. Although future LARP1 structure/function studies – including the region important for mTORC1 binding and LARP1 phosphorylation sites – may shed additional mechanistic insights, the phosphorylation of LARP1 likely induces conformational changes that reduce LARP1’s affinity for the 5’UTR of LARP1- interacting 5’TOP mRNAs. Simultaneously, mTOR- dependent LARP1 phosphorylation may allow LARP1 to serve as a scaffold for mTORC1 on translationally- competent mRNAs. Such a mechanism would support mTORC1-dependent formation of the eIF4F initiation complex, where eIF4G outcompetes LARP1 for interacting with the 5’UTR, as recent studies have proposed (Lahr. et al. *eLife* 2017, Fonseca et al. JBC 2015). Additional studies will obviously be required to understand the precise molecular mechanisms underlying phosphorylation-dependent alterations of LARP1-RP mRNA configuration.

In response to the reviewer’s questions, we examined whether LARP1 dimerizes and found that HA-LARP1 interacts with MYC-LARP1 in vivoin a manner independent of LARP1’s RNA binding (lane 4) or cellular mTOR activity (lane 3), indicating that LARP1 indeed forms at least a homodimer (Figure 8).

Author response image 1.LARP1 forms at least a dimer.HEK293T cells co-transfected with HA- and MYC-LARP1 were treated with Torin1 for 1 hour before harvest. The lysate was treated with RNAseA prior to IP (lane 4).**DOI:**
http://dx.doi.org/10.7554/eLife.25237.020

In addition, our new biochemical data (new Figure 3) suggest that each LARP1 molecule binds its substrate mRNA with a 1:1 stoichiometry: Our new PAR-CLIP assays show that, under cross- linking conditions, LARP1 can bind the wild type 5’/3’ or the mutant 5’conv/3’ RpL32 mRNA equally well in the presence or absence of Torin1 treatment, respectively. These observations suggest that the LARP1 that binds to the wild type 5’/3’ reporter can also bind the 3’ UTR of the 5’conv/3’UTR reporter, even under low mTOR activity conditions.

Taken together, we considered two models of LARP1 interaction with its substrate mRNA. First, in the “Bridging Model,” one LARP1 molecule of the homodimer binds the 5’ UTR, while the other LARP1 molecule of the homodimer binds the 3’UTR of a single RpL32 mRNA (see Figure 9, top left panels). In this configuration, one LARP1 homodimer would bind *both* the 5’ and 3’ of one RpL32 mRNA and “bridge” these two ends of the mRNA. In our luciferase assay, both the 5’ and 3’ UTR of the reporter would be bound by LARP1 in low mTOR conditions. In the Bridging Model, we would predict that the ratio of luciferase mRNA bound to LARP1 homodimer to be 1-to-1. In the alternative, “non-Bridging Model,” LARP1 homodimer would bind *either* the 5’ or 3’ UTR of RpL32, such that a single RpL32 mRNA would be bound by one LARP1 homodimer at its 5’UTR and another homodimer at its 3’UTR. Here, we would predict that the ratio of luciferase mRNA and LARP1 to be 0.5-to-1, i.e. for every mRNA there would be two LARP1 homodimers bound.

Author response image 2.A Hypothetical model for LARP1-RpL32 configuration.The data presented in this study support “One unit LARP1-one mRNA bridging” configuration under low mTOR activity conditions (left).**DOI:**
http://dx.doi.org/10.7554/eLife.25237.021

Our luciferase assays support the Bridging Model: we find that in cells treated with Torin1 as shown in the model (Figure 9), we always find the ratio of luciferase mRNA to LARP1 homodimer to be a 1-to-1 correlation.

As we discussed in our original manuscript, single-molecule visualization or/and more detailed stoichiometric analysis of LARP1 substrate binding will be necessary to fully test the Bridging Model. Nevertheless, our current biochemical data suggest a model where LARP1 is likely bridging the 5’UTR and 3’UTR of the same RpL32 mRNA.

Finally, in response to the reviewer’s question regarding the phospho-mimetic LARP1 mutant, we have substituted serine and threonine phosphorylation sites to aspartic and glutamic acids, respectively (LARP1 3D-E mutant). However, the phospho-mimetic mutant (3DE) behaved the same way as the phospho-defective mutant (4A): the LARP1 3DE mutant displayed reduced binding to mTORC1 much like the LARP1 4A mutant (Figure 10). This is unfortunate but a common phenotype of phospho-defective and mimetic mutants as, in many reported cases, substitutions of S/T phosphorylation sites with aspartic or glutamic acid do not mimic the functions of phosphorylated residues.

Author response image 3.Wild type but not phopho-defectitve or phospho-mimetic LARP1 interacts with mTOR in a rapamycin sensitive manner.**DOI:**
http://dx.doi.org/10.7554/eLife.25237.022